# Lateral transduction is inherent to the life cycle of the archetypical *Salmonella* phage P22

Alfred Fillol-Salom [1,2,6], Rodrigo Bacigalupe [3,4,6], Suzanne Humphrey [1], Yin Ning Chiang [5], John Chen [5✉] & José R. Penadés [1,2,3✉]

Lysogenic induction ends the stable association between a bacteriophage and its host, and the transition to the lytic cycle begins with early prophage excision followed by DNA replication and packaging (ERP). This temporal program is considered universal for P22-like temperate phages, though there is no direct evidence to support the timing and sequence of these events. Here we report that the long-standing ERP program is an observation of the experimentally favored *Salmonella* phage P22 tsc$_2$29 heat-inducible mutant, and that wild-type P22 actually follows the replication-packaging-excision (RPE) program. We find that P22 tsc$_2$29 excises early after induction, but P22 delays excision to just before it is detrimental to phage production. This allows P22 to engage in lateral transduction. Thus, at minimal expense to itself, P22 has tuned the timing of excision to balance propagation with lateral transduction, powering the evolution of its host through gene transfer in the interest of self-preservation.

[1] Institute of Infection, Immunity and Inflammation, College of Medical, Veterinary and Life Sciences, University of Glasgow, Glasgow G12 8TA, UK. [2] MRC Centre for Molecular Bacteriology and Infection, Imperial College London, London SW7 2AZ, UK. [3] Dep. Ciencias Biomédicas, Universidad CEU Cardenal Herrera, 46113 Moncada, Spain. [4] The Rega Institute for Medical Research, KU Leuven, 3000 Leuven, Belgium. [5] Department of Microbiology and Immunology, Infectious Diseases Translational Research Programme, Yong Loo Lin School of Medicine, National University of Singapore, 5 Science Drive 2, Singapore, Singapore. [6] These authors contributed equally: Alfred Fillol-Salom, Rodrigo Bacigalupe. ✉email: miccjy@nus.edu.sg; j.penades@imperial.ac.uk

Bacteriophages (phages) are viruses that infect and replicate within bacteria. They are the most diverse and abundant biological entities on the planet, and they play an important role in controlling populations of bacteria that they exploit and kill. Yet their existence is tied to the success and survival of the hosts that they parasitize, and in the interest of self-preservation, phages can act as powerful agents of microbial adaptation and evolution by means of their ability to transfer bacterial DNA (encoding virulence factors and antibiotic resistance) from one bacterium to another by a process known as genetic transduction.

Phages are obligate intracellular parasites that follow two different life cycles. In the lysogenic cycle, although some are mantained as episomes, most temperate phages reproduce by integrating their genomes into the bacterial host chromosome where they persist in a dormant state as a prophage and replicate passively as part of the bacterial chromosome during cell division[1]. Infectious phage particles are produced in the lytic cycle, following host cell infection or prophage induction from the lysogenic cycle. In most of the archetypical studied viruses, including the *Salmonella enterica* P22 or the *Escherichia coli* λ phages, viral DNA replication first occurs bidirectionally, followed by a switch to rolling-circle replication that generates long DNA concatemers of multiple phage genomes[2,3]. The concatemeric DNA is packaged by the phage terminase enzyme (a hetero-oligomer of small and large terminase proteins), which recognizes a phage-specific packaging site (*pac* or *cos*) and cleaves the DNA to begin processing the genome into newly formed phage heads[4,5]. To complete DNA packaging, *pac*-type terminases make a nonspecific terminal cut when a capsid headful (slightly longer than a genome unit length) has been reached[6,7], while *cos*-type terminases require a second *cos* site to make the second cut[8], forming infectious phage particles that are released during bacterial cell lysis. It is also during this process that transducing particles are formed, when bacterial DNA is packaged into phage heads.

Lysogeny is generally very stable when maintained by sufficient levels of phage repressor, but prophages can switch to the lytic cycle when their hosts begin to deteriorate. They usually do so by exploiting the bacterial SOS response, although additional mechanisms of prophage induction have been recently reported[9]. The SOS response is a global repair pathway that responds to DNA damage and serves as a proxy for the health of the host. Under normal conditions, the SOS-inducible genes are negatively regulated by the bacterial LexA repressor. After a cell experiences DNA damage, the bacterial RecA protein becomes activated by single-stranded DNA and acts as a co-protease to stimulate the self-cleavage of LexA[10,11]. This response is rapid and reversible, and complete LexA cleavage occurs in less than 5 min so that minor insults to the cell can be repaired in a timely manner[12]. Many phage repressors, such as those of phages λ and P22, mimic the structure of LexA and are also induced to undergo self-cleavage by activated RecA[13]. However, phage repressors are cleaved far slower than LexA and take up to 30 min or longer to reach levels that are low enough to de-repress phage lytic genes[12,14,15]. Presumably, this is to ensure that lysogenic induction only occurs in cells that have received prolonged damaging exposure and are not expected to survive. Alternatively, the SOS response can be bypassed entirely with prophages that carry temperature-sensitive repressor (ts*c*) mutations. For instance, thermal induction of λ and P22 ts*c* mutants was the experimental method of choice for prophage de-repression because it occurred within minutes and the cultures were regarded as more synchronized[3,16].

After lysogenic induction, λ- and P22-like prophages are presumed to excise (and circularize) early as the first step in the temporal program of excision–replication–packaging (the ERP cycle; Fig. 1). This sequence of events is widely accepted and believed to be universal for these type of phages because it was thought that DNA packaging prior to excision would split the viral genome and prevent the production of infectious phage particles. Also, experimental support from λ and P22 studies showed that the proteins involved in excision (integrase and excisionase) were early gene products that appeared soon after phage infection or thermal induction of ts*c* mutants[3,16–18]. However, that appears to be the extent of the data on excision timing, and there seems to be no direct evidence that validates the model for early excision[19]. Further complicating the matter, recently we found that some staphylococcal prophages delay excision until later in their lytic cycle[20], and others have shown that some *Salmonella* prophages likely do the same[21]. Therefore, the timing of prophage excision turns out to be an unresolved part of what was thought to be a well-defined phage life cycle. While the distinction between early or delayed prophage excision might seem minor, it has profound evolutionary consequences because only the latter is predicted to result in lateral transduction[20].

Phage-mediated gene transfer is known to occur by three different mechanisms: specialized (ST), generalized (GT), and lateral transduction (LT). In the first, specialized-transducing particles are formed when viral and bacterial DNA hybrid molecules are encapsidated following aberrant prophage excision events[22]. The mechanism is considered specialized because it is limited to the transfer of bacterial genes immediately adjacent to the integrated prophage. In the second, generalized-transducing particles are formed when *pac*-type terminases recognize *pac* site homologs in host DNA and initiate packaging by the headful mechanism[23]. The mechanism is regarded as generalized because any bacterial DNA can be packaged and transferred in this manner. In the recently discovered LT, transducing particles are formed when DNA packaging initiates from the bona fide *pac* sites of prophages that have delayed excision and are still attached to the bacterial chromosome[20,24]. The headful mechanism fills the first capsid with part of the prophage genome and continues through the adjacent bacterial chromosome for up to seven or more successive capsid headfuls. To prevent splitting the integrated viral genome in two, bidirectional in situ replication (prior to DNA packaging) creates sufficient genomic redundancy to enable LT and phage maturation to proceed in parallel. Together, this results in a mode of transduction that produces normal phage titers and transfers bacterial chromosomal DNA at frequencies far greater than previously observed for known mechanisms of host gene transfer. Importantly, LT was described using *S. aureus* phages, raising the question whether this mechanism of gene transfer is exclusive to this species or widespread in nature.

Much of our current understanding of fundamental phage biology derives from the body of work on the archetypal *Salmonella pac*-type P22 and the *E. coli cos*-type λ phages. In those studies, λ or P22 ts*c* mutants were used for lysogenic induction because they were otherwise regarded as WT. Here we found that P22 follows two different temporal programs to produce infectious phage particles, depending on the method of lysogenic induction. Thermal induction of a P22 ts*c* mutant resulted in early prophage excision and the classical ERP program. SOS induction of P22 resulted in delayed prophage excision, DNA replication and packaging in situ, and LT (see scheme in Fig. 1). These results propose a new series of events in the life cycle of P22-like temperate prophages and show that early excision was an artifactual consequence of an experimental system and that delayed excision leading to LT are naturally parts of the phage life cycle.

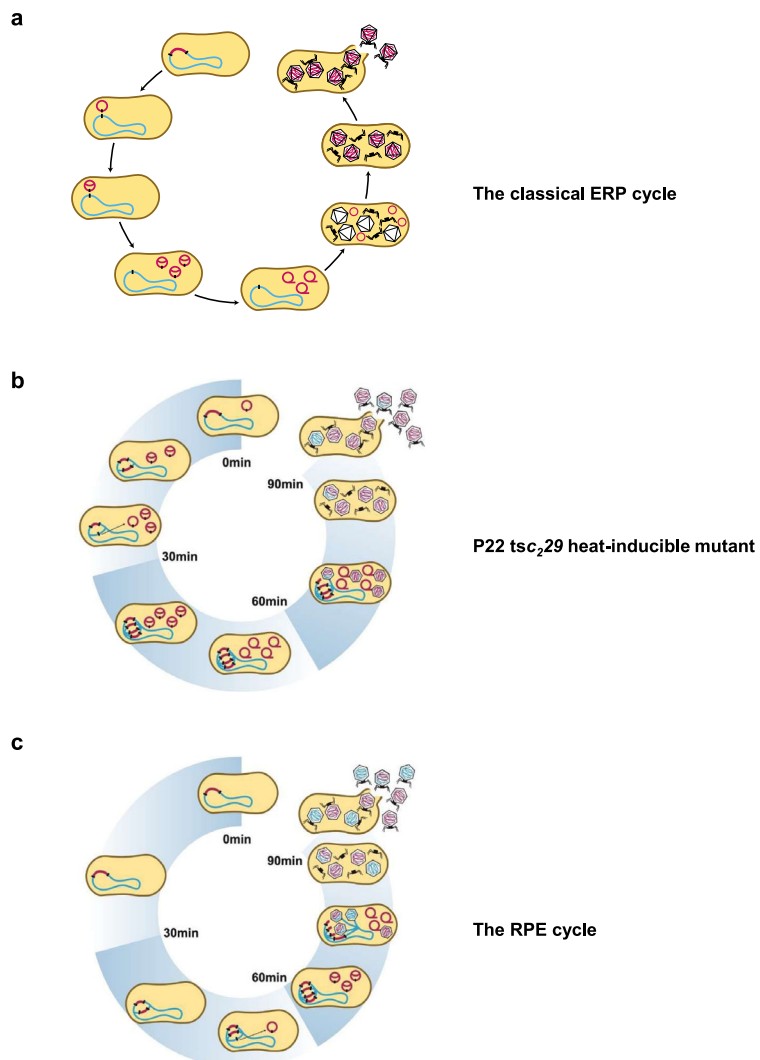

**Fig. 1 The life cycle of temperate prophages after induction. a** Representation of the classical ERP (excision, replication and packaging) model for prophage induction. **b** The P22 ts$c_2$29 heat-inducible mutant follows the ERP cycle. **c** Representation of the RPE (replication, packaging and excision) model. In red, phage genome; in blue, chromosomal DNA.

a

The classical ERP cycle

b

P22 ts$c_2$29 heat-inducible mutant

0min
90min
30min
60min

c

The RPE cycle

0min
90min
30min
60min

## Results

**Lysogenic induction of the P22 or P22 ts$c_2$29 prophages**. As we noted earlier, prophage de-repression by thermal induction was known to occur much more rapidly than SOS induction[3,16]. This disparity in timing suggested that the heat-induced temporal program was either intact (but shifted ahead of schedule) or it was artificial. To distinguish these possibilities, we performed transcriptional profiling of P22 and P22 ts$c_2$29[25]. For phage infection, a non-lysogenic derivative of *Salmonella enterica* LT2 that was cured of all four of its resident prophages was infected with P22. For prophage induction, lysogenic strains were grown at permissive temperature (32 °C) and for thermal induction the P22 ts$c_2$29 lysogen was shifted to 42 °C or to trigger the SOS response the P22 lysogen was treated with mitomycin C. Following thermal induction (for 30 min), the cultures were returned to 32 °C because P22 (WT and ts$c_2$29) forms mostly particles that lack functional tails when incubated at high temperatures for prolonged periods[26]. Total RNA was harvested at 0, 30, 60, and 90 min after infection or induction for RNA-sequencing analysis. We found that following SOS induction of the P22 prophage, most viral transcripts did not appear until 30 to 60 min and only reached maximal levels by 90 min, which was consistent with a gradation effect caused by the inactivation of the P22 repressor over time. In contrast, total viral transcripts reached maximal levels early (before 30 min) following infection, due to the total lack of repressor in naïve cells (Fig. 2). Surprisingly, thermal induction of P22 ts$c_2$29 also resulted in maximal viral RNA levels before 30 min and mirrored the RNA transcript profiles of lytic infection, suggesting that the pool of P22 ts$c_2$29 repressor was inactivated so completely and rapidly that the viral genome was de-regulated to the extent as it was during lytic infection. In fact, this prophage showed higher levels of *xis* and *int* expression even before induction (Fig. 2), suggesting that the c2 C59R mutant (present in P22 ts$c_2$29) is less capable of repressing transcription than WT, which ultimately affects prophage activation and the normal growth of the P22 ts$c_2$29 lysogen. Therefore, compared to SOS induction of P22, the P22 ts$c_2$29 program was shifted early, compacted to a small window of time, and no longer temporally regulated.

In the bottom strand, RNA transcripts of the integrase and excisionase genes (*int* and *xis*) appeared early (before 30 min) after P22 infection or thermal induction of P22 ts$c_2$29, but much later (between 30 and 60 mins) after SOS induction of P22 (Fig. 2; Fig. S1). These results were consistent with the P22 ts$c_2$29

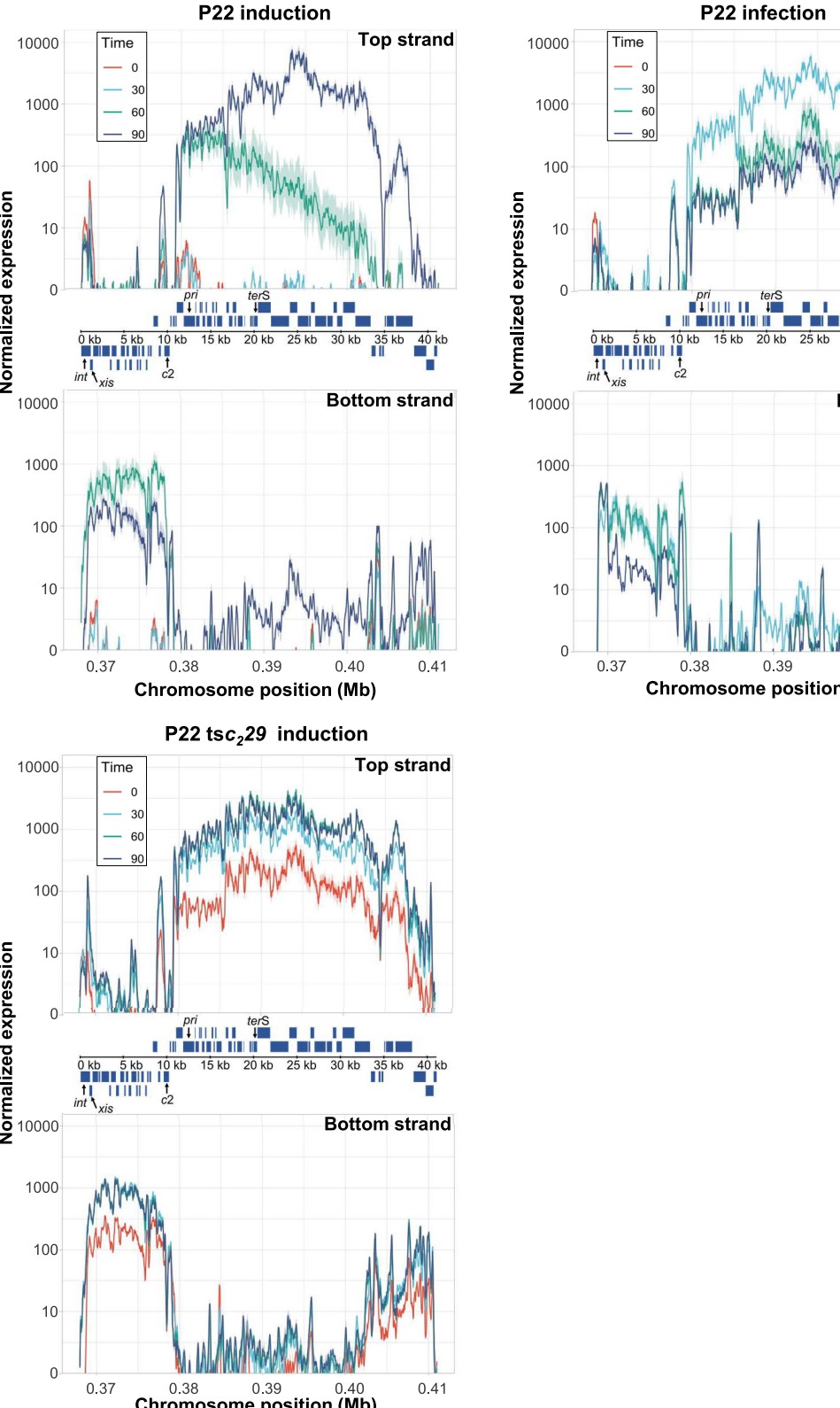

**Fig. 2 Transcriptomic profile of P22 after prophage induction or phage infection.** Expression analysis of the early, middle and late genes from phages P22 and P22 ts$c_2$29 for the positive (top) and the negative (bottom) DNA strands. For analysis of phage induction, lysogenic strains carrying phage P22 or P22 ts$c_2$29 were induced (using mitomycin C or temperature-shift, respectively). For analysis of phage infection, the non-lysogenic LT2 strain was infected with P22. Samples were analyzed without induction or before infection (time 0, red), or at 30 min (early genes, cyan), 60 min (green) or 90 min (late genes, blue) following prophage induction or phage infection. Experiments were performed in triplicate. Shading indicates standard deviation. Expression was normalized by the number of aligned reads.

literature and suggested that P22 ts$c_2$29 excises much earlier than P22 following induction. To correlate the appearance of *int* and *xis* transcripts with prophage excision, the lysogenic strains were induced under the same conditions as above and total chromosomal DNA was isolated for whole-genome sequencing. Here we included ES18[27], a *Salmonella pac*-type phage with an unrelated DNA-packaging module, to broaden the scope of our study. For calculating the integration rates, we first counted the number of sequencing reads that spanned empty prophage attachment sites (*att*B), referred to as chromosome-*att*B-chromosome reads; and the reads covering the left end of the integrated prophage (*att*L site), which represent reads covering the chromosome-*att*L region. At each time point, we compared the number of sequencing reads covering the left end of the integrated prophage (*att*L) with the total number of reads mapping either to the *att*B (empty) or the *att*L (integrated prophage) site (see Fig. S2 for as schematic representation of the stategy used). The resulting value represents the percentage of integrated prophage.

Prior to induction, spontaneous excision was observed for P22 ts$c_2$29 (Fig. S2), owing to the general leakiness of transcription in the mutant (Fig. 2). Following lysogenic induction, the percentage of integrated P22 ts$c_2$29 declined before 30 min, while those of P22 and ES18 remained steady before decreasing between 30 to 60 mins (Fig. S2). These results were consistent with the transcriptomics data and confirmed that lysogenic induction by temperature shift or the SOS response results in early or delayed excision, respectively (Fig. S2). These differences in prophage induction were also evident when we analysed the production of the infective phage particles after induction of the different prophages (Fig. S3).

**Thermal induction results in an artificial temporal program**. Though the percentage of integrated P22 decreased from 30 to 60 mins after SOS induction, it began to change course and increased after 60 min (Fig. S2). This curious result suggested that the P22 prophage was either reintegrating or replicating prior to excision. Previously, we had a similar observation with staphyloccocal prophages and found that in situ replication caused the late increase in the percentage of integrated prophage[20], so we constructed a deletion mutant of the P22 *pri* gene that initiates bidirectional (theta) replication and tested it for excision as above. Shown in Figure S2, the percentage of integrated P22 (Δ*pri*) now decreased from 60 to 90 mins, confirming that P22 excision occurs late and that P22 replicates before excision.

To confirm that P22 replicates in situ, we checked for escape replication, which is a phenomenon that occurs when prophage replication starts before excision and amplifies the bacterial genome flanking the integrated prophage. For example, P22 prophage mutants that are "locked-in" (unable to excise) have been shown to initiate replication in situ and amplify neighboring regions of the chromosome[28]. To test for this, we first deleted the *int* or *xis* genes to generate P22 mutants that are unable to excise to serve as controls for escape replication. Then we infected the non-lysogenic strain with P22 or induced the lysogenic derivatives of P22 (WT, Δ*pri*, Δ*int*, and Δ*xis*) with mitomycin C or P22 ts$c_2$29 with thermal induction and collected the total chromosomal DNA for whole-genome sequencing. At each time point, we quantified the reads corresponding to P22 and the host DNA adjacent to the *att*B$_{P22}$ site and represented them as the coverage relative to the average of the entire genome. P22 infection showed early (before 30 mins) and strong amplification of phage DNA (Fig. S4), which was consistent with the P22 literature[3,16–18]. SOS induction of the P22 lysogenic derivatives

showed clear amplification of phage DNA later in the lytic cycle (Fig. 3), except for the P22 (Δ*pri*) mutant, which remained flat at all time points because it was unable to replicate.

P22 DNA amplification initiated between 30 to 60 mins and was much more robust by 90 mins when episomal replication occurred. By comparison, the P22 (Δ*int*) and P22 (Δ*xis*) mutants showed similar levels of phage DNA amplification at early times, but not at 90 mins because they were not capable of episomal replication. Consistent with their inability to excise, amplification of the flanking bacterial DNA was higher for P22 (Δ*int*) and P22 (Δ*xis*) than P22 by 90 min (Fig. 3), although the phage titers of the *int* and *xis* mutant were significantly reduced (Fig. S3). These results confirm that P22 initiated replication while integrated and amplified the adjacent host DNA.

Compared to SOS induction of P22, thermal induction of the P22 ts$c_2$29 mutant resulted in an altered replication pattern with much greater levels of phage DNA amplification at 0 and 30 min. It is likely that the higher levels of phage DNA amplification observed were due to early-onset episomal replication of the spontaneously-excised genomes (Fig. 3; Fig. 1), which also correlates to higher titers of the phages at the different time points analyzed (Fig. S3). Interestingly, the P22 ts$c_2$29 mutant also amplified flanking bacterial DNA earlier (before 30 min) than P22 (Fig. 3), which was probably due to rapid de-repression of the entire P22 ts$c_2$29 genome from thermal induction (Fig. 2).

To clearly confirm the previous results, we analyzed excision and circularization of the different prophages after MC induction, using qPCR (Fig. S5). Taken together, all our results show that thermal induction resulted in a program where the prophage genome separated early from the bacterial chromosome before replication, while SOS induction resulted in delayed prophage excision and an RPE program. Similar results were also observed for the ES18 prophage (Fig. 3, Fig. S5), indicating that other *Salmonella* prophages initiate replication in situ as well. Therefore, the heat-induced temporal program was not just early, but it is artifactual as well.

**P22-mediated lateral transduction by SOS induction**. P22 is not known to engage in LT. Early studies that characterized theoretical modes of phage transduction found that transductions with lysates from P22 ts$c_2$29 lysogens (that were heat-induced for longer than 75 min) did not support the models that most resembled LT[29]. The main processes of LT are late prophage excision that link to DNA replication and packaging in situ. Since thermal and SOS induction cause very different temporal programs, we reasoned that they would result in equally different outcomes in terms of in situ packaging and LT. To test for in situ DNA packaging, we inserted tetracycline resistance markers (*tet*A) on both sides of the P22 *att*B site, in the directionality of phage packaging (*tet*A$^F$) and in the opposite direction (*tet*A$^R$), to serve as proxies for bacterial gene transfer. The locations of the markers were chosen in reference to the P22 *pac* site, which is embedded in the center of the small terminase gene (*ter*S) and directs packaging unidirectionally toward the 3′ end[5,30]. The headful packaging mechanism reaches capacity (~105% of the genome unit length) around 44 kb, and the markers were placed so that there would be sufficient flanking DNA for homologous recombination in the non-lysogenic recipient. Additional markers in the directionality of terminase packaging were also introduced at positions marking successive capsid headfuls to determine how much of the bacterial chromosome could be mobilized from a single lysogen by the highly processive DNA packaging machinery. These markers were also used for ES18 since both phages integrate at the same *att*B site[27]. Non-lysogenic strains

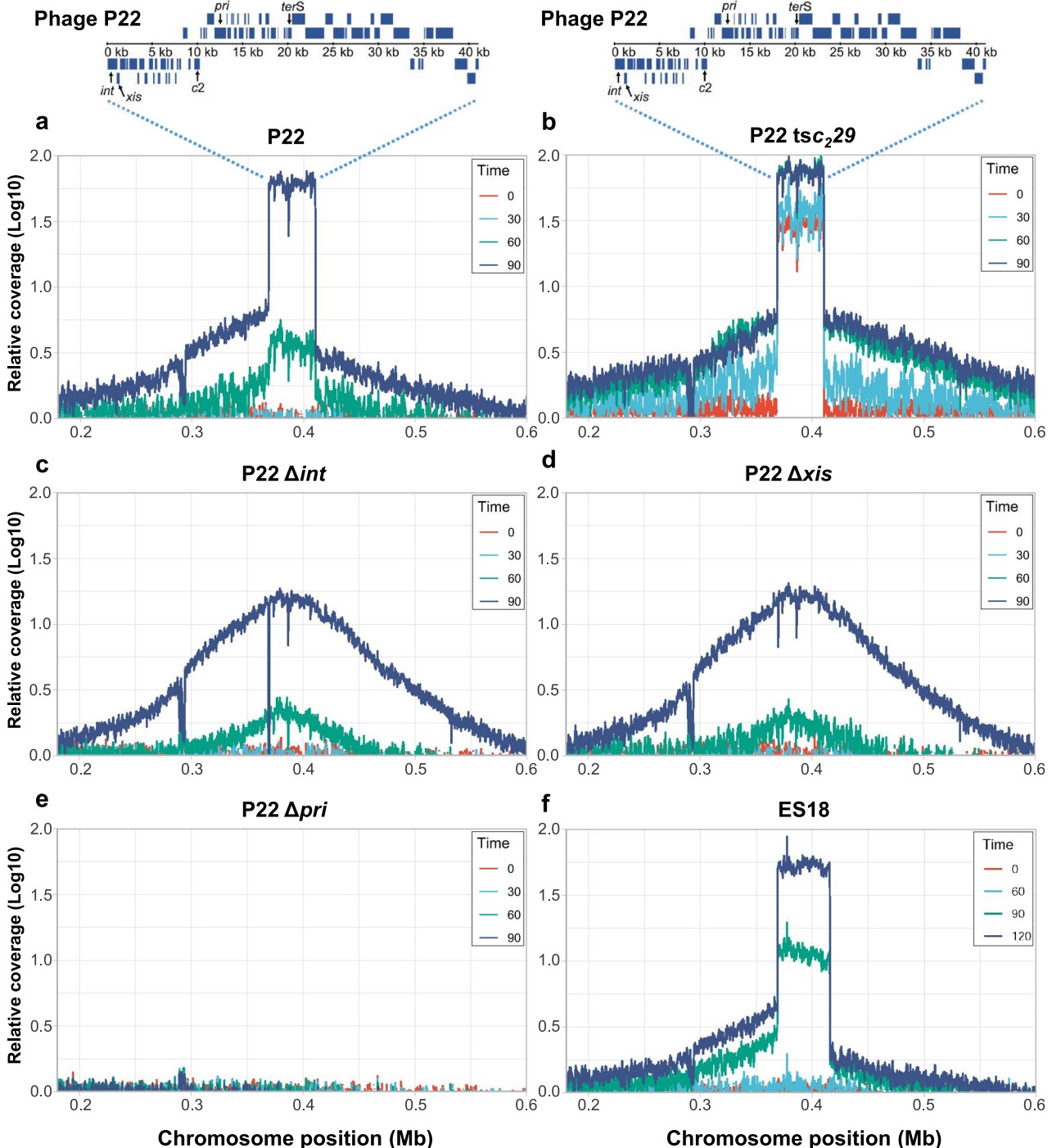

**Fig. 3 Phage P22 and ES18 replicate in situ before excision.** Relative abundance of phage genomic DNA and the chromosomal regions proximal to where they integrate for P22 (**a**), P22 ts$c_2$29 (**b**), P22 Δ*int* (**c**), P22 Δ*xis* (**d**), P22 Δ*pri* (**e**) or ES18 (**f**) is represented. Samples were analyzed at 0 (red), 30 (cyan), 60 (green), and 90 min (blue) after induction of P22 prophage or at 0 (red), 60 (cyan), 90 (green), and 120 min (blue) after induction of ES18 (using mitomycin C or temperature-shift, as appropriate). Relative coverage is the DNA relative to the average bacterial genomic coverage (excluding phages).

(deleted for the P22 *att*B site) carrying these markers were infected with P22, P22 ts$c_2$29 and ES18 to assess GT or strains lysogenic for P22 ts$c_2$29, P22 and ES18 were induced to assess LT and the resulting lysates were tested as donors of tetracycline resistance to *S. enterica*.

We found that after SOS induction of the P22 lysogens, up to seven headfuls (>300 kb) were transferred at frequencies 2-3 orders of magnitude greater than by GT and remained greater than GT for up to twelve headfuls (>500 kb) (Fig. 4a). This P22-mediated LT

occurred by the headful mechanism during lysogenic induction (Fig. S6), rather than by the superinfection of lysogenic strains by phages that were released early after SOS induction (Fig. S7). Importantly, similar results were also observed for ES18-mediated LT (Fig. 4b), suggesting that LT is widespread in *Salmonella* phages, as previously described in *S. aureus* phages[20]. While WT P22 and the P22 ts$c_2$29 mutant showed similar efficiencies in mobilizing markers by GT, the LT frequencies of the first two *tet*A^F markers were significantly lower after thermal induction of P22 ts$c_2$29 than

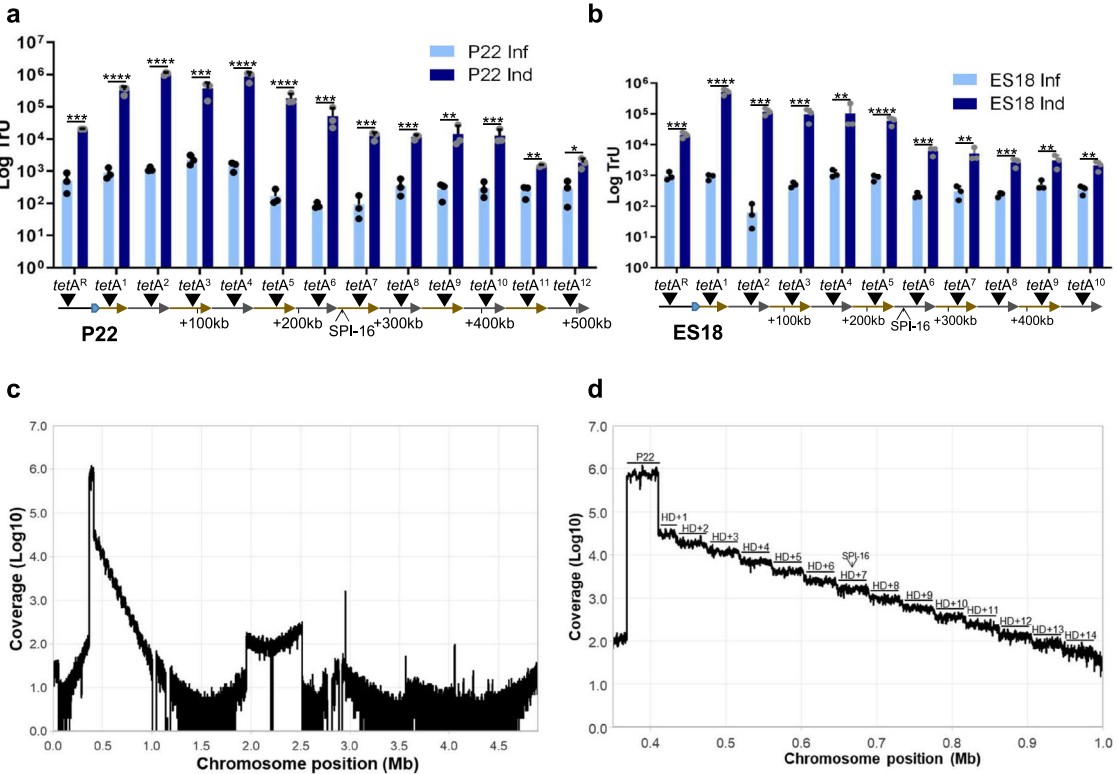

**Fig. 4 P22 and ES18 engage in lateral transduction, transferring large metameric spans of the bacterial chromosome at high frequencies. a** The transfer of tetracycline (*tet*A) markers located upstream (*tet*A$^R$) or downstream of the P22 *att*B site, in twelve successive capsid headfuls (*tet*A$^{"n"}$), was tested following P22 prophage induction (Ind) or P22 infection (Inf). **b** The transfer of *tet*A markers located upstream (*tet*A$^R$) or downstream of the P22 *att*B site, in ten successive capsid headfuls (*tet*A$^{"n"}$), was tested after ES18 induction or ES18 infection. **c** A P22 lysogen was mitomycin C-induced and the resulting phage particles purified. The DNA from the phage particles was extracted and sequenced. The coverage of chromosomal DNA is represented. **d** Zoom of the region encompassing lateral transduction is visualized, displaying a successive 'stepdown' pattern in DNA packaging efficiency for each consecutive headful. Transduction units (TrU) per milliliter were normalized by PFU per milliliter and represented as the log TrU of an average phage titer ($1 \times 10^9$ PFU). Error bars indicate standard deviation from the mean of three independent experiments. For all panels, values are means ($n = 3$ independent samples). An unpaired two-sided *t* test was performed to compare mean differences between infection and induction in each marker. Adjusted *p* values were as follows: ns > 0.05; \**p* ≤ 0.05; \*\**p* ≤ 0.01; \*\*\**p* ≤ 0.001; \*\*\*\**p* ≤ 0.0001. The exact statistical values for each of the conditions tested are listed on Table S1.

SOS induction of P22 (Fig. S8). This result likely explains why earlier studies did not observe LT with the P22 ts$c_2$29 mutant. Taken together, these results show that P22 and ES18 engage in LT and transmit a large section of the *Salmonella* chromosome at high frequencies during SOS (but not thermal) induction. To expand our studies even further, we tested for LT in an *Enterococcus faecalis* lysogen of phage pp1[31]. As shown in Fig. S9, following SOS induction, prophage pp1 transferred *tet*M markers in the directionality of packaging at much higher frequencies than a marker in an unlinked location in the chromosome, showing that LT occurs in *E. faecalis* and it is a widespread and universal mechanism of gene transfer.

For a more direct visualization of LT, we purified P22 particles resulting from SOS induction and extracted the DNA for sequencing. The reads were quantified and mapped to the bacterial chromosome (NC_003197) and quantified based on coverage. We found that 96.5% of the reads of bacterial origin mapped to a 590 kb region in the directionality of packaging by P22-mediated LT. The remaining reads of bacterial origin were randomly distributed along the whole genome, which was indicative of GT (Fig. 4c). The individual headfuls could be visualized by a step down in DNA packaging efficiency for each successive headful, corresponding to each re-initiation of DNA packaging by the processive terminase enzyme (Fig. 4d). Each step was approximately 44 kb in length, which matched the predicted headful capacity for P22.

**The timing of prophage excision is tuned for lateral transduction in the newly identified P22 RPE cycle.** Because early excision (by the P22 ts$c_2$29 mutant) resulted in high phage titers (Fig. S3) and low levels of LT (Fig. S8), the opposite should be true for very late excision. This was indeed the case, as the P22 (Δ*int*) and P22 (Δ*xis*) single mutants or a P22 (Δ*int-xis*) double mutant resulted in much lower phage titers and higher levels of LT than WT P22 (Fig. S10). Matching results were obtained when different ES18 mutants were also analyzed (Fig. S10). Therefore, we wondered if there was a cost to P22 for delayed excision in terms of phage production. To test this, we designed a system in which prophage excision was tightly regulated and inducible. The P22 *int-xis* double mutant was complemented with a plasmid expressing the *int* and *xis* genes under the control of an arabinose-inducible promoter. To test for LT, we used the *tet*A$^F$ marker located in the second P22 headful, thus avoiding any possible interference with ST. This strain was SOS-induced with mitomycin C [time (t) = 0 min] and arabinose was added at different time points ranging from 0 to 4 h (after mitomycin C) for *int-xis* expression. As expected, early expression (t = 0 mins) of the *int* and *xis* genes after SOS induction produced high phage titers but significantly reduced LT, while very late expression (t = 240 mins) resulted in low phage titers and high levels of LT (Fig. 5). Interestingly, *int-xis* expression at 0 to 60 mins maintained P22 titers at the same high levels, with only a minor

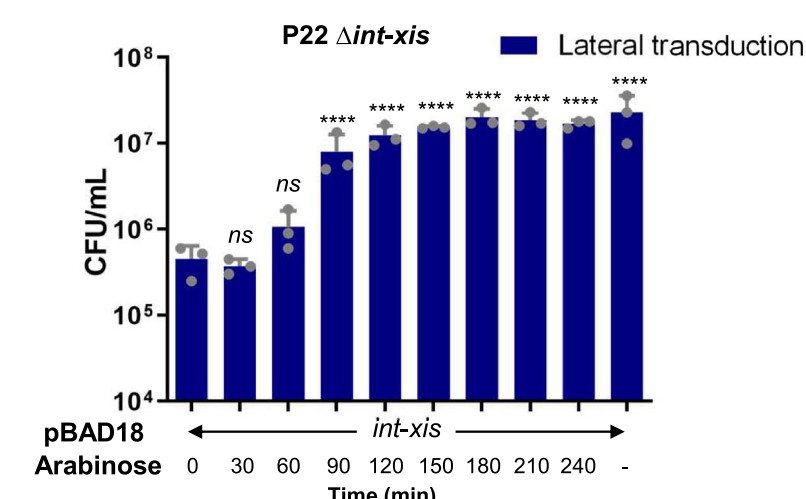

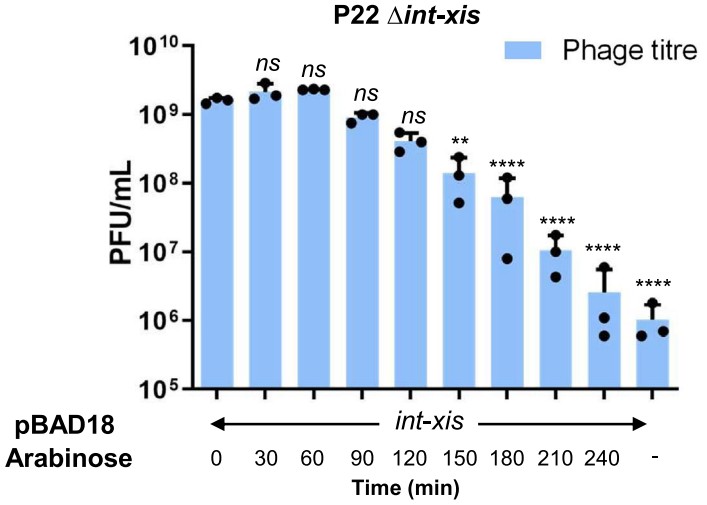

**Fig. 5 Effect of early or delayed excision on lateral transduction and phage formation.** The lysogenic strain for phage P22 mutant Δ*int-xis*, carrying a plasmid expressing the *int* and *xis* genes, was mitomycin C-induced to activate the prophage, and arabinose (0.02%) was added at different time points after induction of the resident prophage to facilitate *int-xis* expression *in trans*. (−): no arabinose added. Lysates were tested for transduction of the tetracycline (*tet*A) marker (colony forming units per millilitre: CFU/mL) (**a**) and to determine the phage titers (plaque forming units per millilitre: PFU/mL) (**b**). Error bars indicate standard deviation from the mean of three independent experiments. For all panels, values are means ($n = 3$ independent samples). A one-way ANOVA with Dunnett's multiple comparisons test was performed to compare time 0 against the other timepoints. Adjusted *p* values were as follows: *ns* > 0.05; **p* ≤ 0.05; ***p* ≤ 0.01; ****p* ≤ 0.001; *****p* ≤ 0.0001. The exact statistical values for each of the conditions tested are listed in Table S1.

decline at 90 mins. At 120 to 240 mins, the phage titer declined rapidly, presumably because in situ DNA packaging compromised the integrity of the prophage genomes. By comparison, *int-xis* expression at 90 to 240 mins was optimal for LT. These results indicate that a delay in excision up to 60 to 90 min imposes a minimal cost to phage production and allows P22 to engage in LT. Thus, P22 prophage excision occurs in a narrow window of time that balances maximum phage production with LT.

## Discussion
Phages have traditionally been viewed as selfish elements that exploit bacteria for their reproduction and spread in nature. The ERP program underscored this self-serving lifestyle because early

prophage excision severed the connection to the bacterial cell and opened the way for unchecked viral propagation without any regard for the host. This sequence of events was based on the work from early studies with archetypal phages P22 and λ, and it has stood as the life cycle described for many temperate phages following lysogenic induction. However, phages are perhaps not as selfish as once presumed, and here we found that the temperate P22 phage life cycle is much more intimate and mutually beneficial than previously understood. Our results demonstrate, as occurred with the *S. aureus* phages[20], that the P22 temperate phage does not follow the ERP program, but instead follows an RPE program, so that the prophage remains connected to the host chromosome for most of its life cycle following lysogenic

induction. It does this by starting with in situ replication, which generates multiple copies of the integrated prophage to create genomic redundancy. Once the DNA packaging machinery is assembled, in situ packaging initiates from some of the integrated prophages and excision occurs from some of the others, so that the production of lateral-transducing particles occurs during normal phage maturation (see scheme in Fig. 1c).

Lysogenic protection and "morons" are the most common examples of benefits that temperate prophages provide to their bacterial hosts. In the first, prophages provide immunity against infection by other phages encoding identical or similar repressors. Phage morons are accessory genes that are not required for the phage life cycle but can increase the fitness of the lysogenic cell[32]. Importantly, defective prophages can still provide the benefits of lysogenic protection and morons, while prophages must be intact and functional to impart the benefits of LT. Thus, the RPE cycle (including LT) is an evolutionary force that preserves prophage integrity so that they remain intact and able to propagate. Moreover, we also propose here that LT is not just a universal mechanism of gene transfer, but an event that forms an intrinsic part of the life cycle of temperate pac prophages. We have recently demonstrated that phages benefit from GT[33], and based on the frequency of gene transfer associated with this process, we hypothesize that phages benefit even more from LT. Thus, once prophage genomes integrate into the bacterial chromosome, their fates are connected in a mutually beneficial relationship.

An interesting observation from our studies is that both in situ replication and in situ packaging must be perfectly coordinated to allow phages to engage in LT. This was best illustrated with P22 $tsc_229$, where high levels of in situ replication were observed, but not high frequencies of LT, since most of the DNA packaging machinery was probably titrated away by the high levels of epi-somally replicating viral DNA. Likewise, our results also highlight the fine regulation that occurs between in situ packaging and late excision, which balances optimal phage reproduction with LT.

The RPE cycle may not just be an efficient evolutionary strategy that ensures efficient phage reproduction linked to high mobilization and transfer of the bacterial chromosome, but it could also be an interesting mechanism used to overexpress some virulence genes under stress conditions. We have recently demonstrated that many pathogenicity islands are located next to S. aureus and Salmonella phage attB sites[20,34]. One can hypothesize that under stress conditions, the resident prophages in a subpopulation of cells will be induced. As a consequence of in situ replication, the copy number of virulence genes located next to attB sites will also increase, which can significantly augment the expression of particular host proteins. A handful of previous studies have shown that some cos phages, including the proto-typical E. coli λ or the S. enteritidis Fels-1 that are not capable of engaging in GT or LT, replicate in situ before excision[21,35–37]. Although this idea is currently under study, these results suggest that the RPE cycle provides a fitness advantage to the phage and to the host beyond LT.

In summary, the results presented here re-write important concepts of P22 phage biology, including the series of events that occur after lysogenic induction. Moreover, these results also have the potential to fundamentally re-structure our understanding of the roles phages play in bacterial evolution, including the development of antimicrobial resistance and virulence in clinical strains.

## Methods

**Bacterial strains and growth conditions**. Bacterial strains used in this study are listed in Table S2. *Salmonella enterica* strains were grown at 30 °C, 37 °C or 42 °C on Luria-Bertani (LB) agar or in LB broth with shaking (120 r.p.m.). *Enterococcus faecalis* strains were grown at 37 °C on Brain Heart Infusion (BHI) agar or in BHI broth with shaking (120 r.p.m.). For antibiotic selection, ampicillin (100 μg ml⁻¹), kanamycin (30 μg ml⁻¹), chloramphenicol (10 or 20 μg ml⁻¹) or tetracycline (3 or 20 μg ml⁻¹), from Sigma, was added when appropriate.

**DNA methods**. Gene insertions or deletions in *Salmonella* were performed as described[38]. Briefly, the chloramphenicol (*cat*), kanamycin (*km*R) or tetracycline (*tet*A) resistance makers were amplified by PCR, with primers listed in Table S3. PCR products were introduced by transformation into the appropriate recipient strains harboring plasmid pKD46, which expresses the λ Red recombinase, and the markers were inserted in the bacterial or phage genome. The different mutants obtained were verified by PCR. To eliminate the marker, plasmid FLP helper pCP20 was transformed into the strains containing the resistance marker insertions. Strains containing the plasmid pCP20 were grown overnight at 30 °C and a 1:50 dilution (with fresh LB) was prepared and grown for 4 h at 42 °C to encourage plasmid loss and produce FLP recombination. Strains were plated out on LB plates and incubated for 24 h at 37 °C. Individual colonies were streaked out and PCR was performed to corroborate that the chromosomal marker had been removed and were sequenced by Sanger sequencing (Eurofins Genomics).

Insertion of a *tet*M cassette into the *E. faecalis* chromosome was performed by double recombination using allelic exchange plasmid pBT2βgal. Briefly, plasmids pJP2563, pJP2564, pJP2565, pJP2566 and pJP2568 were introduced into *E. faecalis* V583-derivative strains VE18990 (pp-) and VE18562 (φp1) by transformation, with selection at 32 °C on BHI agar supplemented with 10 μg ml⁻¹ chloramphenicol. Following two rounds of sequential incubation at 32 °C and 42 °C, as previously described[39], individual tetracycline-resistant, chloramphenicol-sensitive colonies were selected, and PCR was performed to confirm the correct placement of the tetracycline marker in the bacterial chromosome. Sanger sequencing (Eurofins Genomics) of PCR products corroborated marker placement.

**Plasmid construction**. Plasmid pJP2534 (Table S4) was generated by cloning the PCR product containing the P22 *int-xis* region, amplified with the oligonucleotides listed in Table S3 (Sigma), into the pBAD18 vector. The cloned plasmid was verified by Sanger sequencing (Eurofins Genomics).

For insertion of *tet*M into the *E. faecalis* chromosome, we made use of the thermosensitive allelic exchange plasmid pBT2bgal. Briefly, the *E. faecalis* chromosomal regions flanking the proposed *tet*M insertion site were amplified by PCR using the oligonucleotides listed in Table S3. The *tet*M cassette was amplified from plasmid pRN6680[40] using the oligonucleotides indicated. Overlap extension PCR was used to produce chromosomal left flank-*tet*M fusion products, which were cloned into pBT2bgal alongside the right-flank PCR product, generating plasmids pJP2563, pJP2564, pJP2565, pJP2566 and pJP2568 (Table S4). All plasmid sequences were verified by Sanger sequencing (Eurofins Genomics).

**Induction experiments**. For *Salmonella* phage induction, overnight cultures of lysogenic donor strains were diluted 1:50 in fresh LB broth, then grown at 32 °C and 120 r.p.m. until an $OD_{600}$ of 0.2 was reached. Prophages were induced by either addition of mitomycin C (MC) (2 μg ml⁻¹) or by shifting the temperature to 42 °C (30 min) for thermal induction. Following induction, cultures were incubated at 32 °C with slow shaking (80 r.p.m.). Cell lysis occurred 3–4 h post-induction.

For *E. faecalis* phage induction, overnight cultures of lysogenic donor strains were diluted 1:25 into fresh BHI broth, then grown at 37 °C, 120 r.p.m. until an $OD_{540}$ of 0.10 was reached. Prophages were induced by addition of MC (2 μg ml⁻¹). Following induction, cultures were incubated at 32 °C, 80 r.p.m. for 5.5 h to permit cell lysis. Cell lysis occurred 4-5 h post-induction.

**Infection experiments**. For *Salmonella* phage infection, recipient strains were diluted 1:50 in fresh LB broth and incubated at 32 °C and 120 r.p.m. until an $OD_{650}$ of 0.15 was reached. The culture was then centrifuged for 5 min at 4 °C/3000 × g. The supernatant was discarded, and the pellet re-suspended at a ratio 1:1 of LB-phage buffer (phage buffer [PHB]: 1 mM NaCl, 0.05 M Tris pH 7.8, 1 mM $MgSO_4$, 4 mM $CaCl_2$). The culture was then infected with a phage lysate at a ratio of 1:1 and incubated at 32 °C with slow shaking (80 r.p.m.). Generally, cell lysis occurred 4–5 h post-infection.

For *E. faecalis* phage infection, recipient strains were diluted 1:25 into fresh BHI broth, then grown at 37 °C, 120 r.p.m. until an $OD_{540}$ of 0.34 was reached. 55 μl of recipient cells were mixed with 100 μl phage lysate diluted in PHB (2.0 ×10⁷ Φp1 PFU, yielding an infection ratio of 1:1), and incubated at room temperature for 10 mins before addition of 3 ml phage top agar (PTA: 20 g Nutrient broth no. 2, 3.5 g agar, 10 mM $MgCl_2$). The entire volume was poured onto phage base agar (PBA: 25 g of Nutrient Broth No. 2, Oxoid; 7 g agar, Formedium) supplemented with 10 mM $MgCl_2$, then incubated at 37 °C for 16–18 h. Phage plaques were harvested by collecting and resuspending the PTA layer in 3 ml PHB. Samples were vortexed for 20 s, then centrifuged for 10 min at 4000 × g, and the resulting supernatants were filter sterilized.

Phage lysates obtained either by infection or induction were filtered using sterile 0.2 μm filters (Minisart® single use syringe filter unit, hydrophilic and non-pryogenic, Sartonium Stedim Biotech) and the number of phage or transducing particles quantified.

**Phage titration and transduction**. For *Salmonella* phage titers, serial dilutions of phage lysates were prepared in PHB, and 100 μl of diluted lysate was used to infect 50 μl of cells at $OD_{600}$ 0.34 for 10 min at room temperature. For *E. faecalis* phage titers, serial dilutions of phage lysates were prepared in PHB, and 100 μl of diluted lysate was used to infect 55 μl of VE18590 (pp-) recipient cells at $OD_{540}$ 0.34 for 10 min at room temperature. Following incubation, 3 ml PTA (supplemented to a final concentration of 10 mM with $CaCl_2$, for *Salmonella* phages, or $MgCl_2$, for *E. faecalis* phages) was added to stop the infection process, and the different combinations of culture:phage dilutions were plated out on PBA supplemented to a final concentration of 10 mM with either $CaCl_2$ or $MgCl_2$ as appropriate. PBA plates were incubated at 37 °C for 24 h and the number of plaques quantified as plaque forming units (PFU)/ml.

For transductions, a total of 1 ml of *Salmonella* recipient cells at $OD_{600}$ of 1.4 supplemented with 4.4 mM of $CaCl_2$ or 1 ml of *E. faecalis* recipient cells at $OD_{540}$ of 1.4 supplemented with 4.4 mM of $MgCl_2$, were infected for 30 mins at 37 °C with the addition of 100 μl of phage lysate diluted in PHB. Following incubation, 3 ml LB top-agar (LTA: 20 g LB, Sigma; 7.5 g agar, Formedium) or BHI top-agar (BTA: 37 g BHI, Oxoid; 7.5 g agar, Formedium) was added to stop the transduction process, and the entire contents were poured onto LBA (*Salmonella*) or BHI (*E. faecalis*) agar plates containing the appropriate antibiotic. Plates were incubated at 37 °C for 24 h (*Salmonella*) or 36 h (*E. faecalis*) and the number of colonies formed (transduction particles present in the lysate) were counted and represented as the transduction forming units (TFU)/ml. Results were generally reported as the transduction units ($TrU/10^9$ PFU), which represents the number of transducing particles per 1E + 9 of PFU.

**Co-transduction analyses**. The frequency of marker co-transduction was determined using lysates generated either by phage induction or by phage infection of the strains containing both resistance markers (*cat* and *tet*A), at varying distances apart. 100 *tet*A-resistant transductants were tested for *cat* resistance and the results were represented as the frequency of (*cat/tet*A) x 100%.

**Inducible complementation of the P22 Δ*int-xis* mutant**. The P22 Δ*int-xis* lysogens complemented in trans with the pBAD18 derivative plasmid expressing the *int-xis* genes were diluted 1:50 (in fresh LB broth supplemented with ampicillin at 100 μg ml⁻¹) and were grown at 32 °C, 120 r.p.m. until an $OD_{600}$ of 0.2 was reached. Cultures were induced with MC ($t = 0$) and incubated at 32 °C with slow shaking (80 r.p.m.). Arabinose (0.02%) was added at 0, 30, 60, 90, 120, 150, 180 and 240 min post-induction for complementation of *int-xis* functions. Cultures were filtered using sterile 0.2 μm filters and the number of phage and transducing particles in the resultant lysate were quantified.

**Whole genome sequencing (in situ replication)**. Samples were induced as described in previous sections. At the indicated time points after MC-induction, 12 ml of sample was collected and DNA extraction was performed using the 'ChargeSwitch® gDNA Mini Bacteria Kit' from Thermo Fisher Scientific, following the manufacturer's instructions. The DNA was precipitated by 0.3 M NaOAc and 2.25 volume of 100% ethanol, then pelleted at $12,000 \times g$ for 30 min at 4 °C and washed once with 1 ml of 70% ethanol. After centrifugation, the DNA pellets were air dried for 30 min and resuspended in 50 μl nuclease free water. Quality control of DNA samples was tested using Agilent Bioanalyzer 2100 and whole genome sequencing (WGS) was performed at the University of Glasgow Polyomics Facility using Illumina TruSeq DNA Nano library prep, obtaining 2 × 75 bp pair end reads with DNA PCR free libraries. A total of 2.2 M reads were generated and trimmed reads were mapped to the appropriate genome: P22 (NC_002371.2), ES18 (NC_006949) and LT2 (NC_003197). Only one replicate was sequenced per experiment.

**Whole genome sequencing (phage DNA from capsid extraction)**. A total of 100 ml P22 phage lysate was produced by MC induction. Phage lysates were treated with RNase (1 μg ml⁻¹) and DNase (10 μg ml⁻¹) for 30 min at room temperature. Followed by addition of 1 M of NaCl to the lysate for 1 h on ice. After incubation, the mix was centrifuged at $11,000 \times g$ for 10 min at 4 °C, and phages were resuspended with 10% wt/vol polyethylene glycol (PEG) 8000 and kept overnight at 4 °C. Phages were precipitated at 11,000 x $g$ for 10 min at 4 °C and resuspended in 1 ml of phage buffer. The precipitated phages were loaded on the CsCl step gradients (1.35, 1.5 and 1.7 g ml⁻¹ fractions) and centrifuged at $80,000 \times g$ for 2 h at 4 °C. The phage band was extracted from the CsCl gradients using a 23-gauge needle and syringe. Phages were dialyzed overnight to remove CsCl excess using SnakeSkin™ Dialysis Tubing (3.5 K MWCO, 16 mm dry) into 50 mM of Tris pH 8 and 150 mM NaCl buffer.

Following CsCl purification, phage lysates were treated with DNase (10 μg ml⁻¹) for 30 min at room temperature and DNase was inactivated with 5 mM EDTA for 10 min at 70 °C. Phage lysate was combined with an equal volume of lysis mix (2% SDS and 90 μg ml⁻¹ proteinaseK) and incubated at 55 °C for 1 h. DNA was extracted with an equal volume of phenol:chloroform:isoamyl alcohol 25:24:1 and samples were centrifuged at $12,000 \times g$ for 5 min, and the aqueous phase containing the DNA was obtained. DNA was ethanol precipitated as already described, and the resulting DNA was resuspended in 100 μl nuclease free water.

Quality control of DNA samples was tested using Agilent Bioanalyzer 2100 and WGS was performed at the University of Glasgow Polyomics Facility using Illumina TruSeq DNA Nano library prep, obtaining 2 × 75 bp pair end reads with DNA PCR free libraries. A total of 3000X bacterial genome coverage, 104 M reads, were generated and trimmed reads were mapped to the appropriate genome: P22 (NC_002371.2) and LT2 (NC_003197). Only one replicate was sequenced per experiment.

**WGS analysis**. We first used FastQC v0.11.8 (http://www.bioinformatics.babraham.ac.uk/projects/fastqc/) to assess the quality of the sequencing reads and Trimmomatic v0.36 to remove adapters and low-quality reads. Sequencing reads from each experiment were mapped to their respective reference genomes using the Burrows-Wheeler Alignment Tool 0.7.17 (https://academic.oup.com/bioinformatics/article/25/14/1754/225615). Picard-tools v2.1.1 (http://broadinstitute.github.io/picard/; Broad Institute) was next used to obtain the bam files, which were merged with SAMtools v1.11[41], sorted and indexed; and Bedtools v2.30.0 subcommand *bamtobed*[42] was used to produce the bed files. We computed the relative coverage over 100 sliding windows along the entire chromosome for each of the experiments. For this, we computed the average coverages across the full genome without phages, which were removed using bedtools subcommand *subtract*. Subsequently, the coverages across the sliding windows were divided by the chromosomal averages. In order to estimate the percentage of integration, we calculated the number of reads that spanned the bacterial chromosome and phages on the two ends of the P22 and ES18 phages. In addition, we counted the sequencing reads spanning the empty prophage attachment sites (*att*B) and the reads covering the left end of the integrated prophage (*att*L site). This was performed using the 'view' subcommand of samtools and filtering by the coordinates of the mapping reads and their respective lengths. Next, the integration percentages were obtained by dividing the reads covering the chromosome-*att*L-prophage (chr:pha) by the total reads (chr:pha + chr:chr). Fisher tests on the sequencing read counts (chromosome-*att*B-chromosome and chromosome-*att*L-prophage reads) between different time points (0 min vs 30 min, 30 min vs 60 min, and 60 min vs 90 min) indicate that all apparent changes are statistically significant.

Finally, the sequencing reads of the capsids were rarefied by subsampling 100 million reads from each of the paired, filtered reads using seqtk v1.3 *sample* command (https://github.com/lh3/seqtk). Reads were mapped to the NC_003197 reference genomes and processed as above. The absolute coverage was calculated using bedtools by 100 bp windows across the entire genome. Assemblies of the content of the capsids were obtained using SPAdes v3.12[43]. All the bioinformatic analyses were performed using the Cloud Infrastructure for Microbial Bioinformatics (CLIMB).

**Total RNA extraction and mRNA enrichment**. Following induction of lysogenic strains, 30 ml samples were collected at the indicated time points post-MC addition, and cell pellets were harvested by centrifugation at $5000 \times g$ for 5 min. The bacterial pellet was resuspended in TE buffer (10 mM Tris-Cl, 1 mM EDTA, pH 8.0) containing lysozyme (50 mg ml⁻¹) and was lysed in a FastPrep-24 homogenizer (MP Biomedicals) using two cycles of 60 s at 6.5 m s-2 interrupted by a 5 min incubation on ice. Total RNA was extracted using an RNeasy kit (Qiagen) according to the manufacturer's instructions and genomic DNA was removed using TURBO DNase (Ambion, Carlsbad, CA, USA). RNA samples were enriched for mRNA using MICROBexpress mRNA enrichment kit (Ambion). Quality control of RNA-seq samples for mRNA enrichment was tested using Agilent Bioanalyzer 2100 at the University of Glasgow Polyomics Facility. Experiments were performed in triplicate.

**RNA-seq transcriptome analysis**. cDNA synthesis and sequencing, 75 bp single end reads, was performed at the University of Glasgow Polyomics Facility using Illumina NextSeq 500. The sequencing reads were processed using Trimmomatic v0.36[44] for removal of adapters and low quality reads. Transcriptomic analyses were performed using the pipeline READemption v0.4.3[45]. First, we created individual projects for each experiment using the create subcommand and the fasta and annotation files of the reference genome (P22: NC_002371) were transferred to the corresponding locations. Next, we performed the alignment step using default parameters, and the coverage was computed using only the uniquely aligned reads, normalizing by these reads. The strand specific coverage counts normalized by 100,000 division were used for the final analyses. Strands coverage values of the phage spanning regions for different replicates were plotted with values >1 log10 corrected.

**Quantitative PCR (qPCR)**. Samples were temperature or MC-induced and at the indicated time points, 6 ml of each sample was collected for DNA extraction, as previously indicated. qPCR was performed using the Power Up SYBR Green master mix on the Applied Biosystems StepOnePlus system. Two nanograms of DNA was used per reaction with primers listed in Table S3 at 10 μM final concentration. *rap*A was used as a reference gene.

The relative excision or circularisation levels within distinct experiments were determined with the $2^{-\Delta CT}$ method[46].

**Statistical analyses**. All statistical analyses were performed as indicated in the figure legends using GraphPad Prism 6.01 and R v4.0.3 software. Statistical value for each experiment is listed in Table S1.

**Reporting summary**. Further information on research design is available in the Nature Research Reporting Summary linked to this article.

## Data availability

Source data are provided with this paper. The WGS and RNAseq data generated in this study have been deposited in the NCBI SRA database under accession code BioProject PRJNA737196.

The different accession numbers (obtained from the NCBI database) for reference genomes and the bioinformatics analyses are: LT2 (NC_003197), P22 (NC_002371.2) and ES18 (NC_006949). Source data are provided with this paper.

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

## Acknowledgements

This work was supported in part by the Singapore Ministry of Education MOE2017-T2-2-163 and MOE2019-T2-2-162 to J.C.; and by MR/M003876/1, MR/V000772/1 and MR/S00940X/1 from the Medical Research Council (UK), BB/N002873/1, BB/V002376/1 and BB/S003835/1 from the Biotechnology and Biological Sciences Research Council (BBSRC, UK), ERC-ADG-2014 Proposal nº 670932 Dut-signal (from EU), to J.R.P; and Wellcome Trust 201531/Z/16/Z to J.R.P.

## Author contributions

J.C. and J.R.P. conceived the study. A.F.-S., S.H. and Y.N.C., conducted the experiments. R.B. performed the genomic and transcriptomic analyses. J.C. and J.R.P. wrote the manuscript.

## Competing interests

The authors declare no competing interests.
