## [Peer Review File · Nature Communications]

Lateral transduction is inherent to the life cycle of the archetypical Salmonella phage P22REVIEWER COMMENTS

Reviewer #1 (Remarks to the Author):

Filloi-Salom & Bacigalupe et al examine the canonical excision-replication-packaging paradigm in the well-studied phage P22 and present compelling data supporting the notion that temperate pac type phages engage instead in a replication-excision-packaging program. Such a program lends itself to 'lateral transduction' (LT)– discovered by this group in *Staph aureus*. Demonstrating that LT is not limited to the previously studied staphylococcal phages is of broad interest and significance. I do however think the paper could benefit from being re-framed: the emphasis seems to be on demonstrating that the TS P22 mutant phage is different than WT P22, the latter of which shows 'higher' levels of lateral transduction. This emphasis is likely more interesting/significant to people who study P22, but for a broad audience, this is not really a particularly compelling point. The significance is that the authors show LT in unrelated *Salmonella* phages – suggesting LT is widespread, and lending support to the hypothesis that it is beneficial for phages to engage in LT. For those reasons I suggest the authors more deliberately explain the relevance of including ES18 (and how it differs from P22 specifically), and include the ES18 data in the main manuscript. I would likewise suggest some of the TS P22 mutant data gets moved to the supplement.

Specific comments on figures/analysis follow below:

- 1) Throughout the manuscript it is not clear if replicates were done, and what exactly is being shown as far as replicates and error. This is true for all RNA and DNA-seq data (ie figs 1, 2, 4, S2 ect). The data points for each replicate should also be shown on each bar graph in addition to the error
- 2) It would be helpful to orient the experiments if the authors included a time course showing PFU production for the different strains, this would help put the RNA/DNA-seq in context
- 3) It is clear from the RNA & DNA-seq data that P22 TS mutant is already induced without the temp shift, I have a hard time believing those cells are not sick and producing a significant amount of phage – this makes me concerned about suppressor mutations that could be confounding the results somewhat
- 4) Figure 1 – the authors are using these data to conclude that *int/xis* are not expressed until late in infection in WT P22. However, it is unclear from the RNA-seq data alone that this is the case – this needs to be corroborated by analysis at the protein level. I also suggest that they provide a zoomed in view of the RNA-seq data on the *int/xis* operon – but again, this is not sufficient to conclude anything given that post transcriptional regulation may play a role and they are drawing their conclusions based on this data alone. I would also like to understand a little more about what is encoded downstream of *xis* that would need to be on late in infection, I'm sure there is a lot known and it would be helpful for the general reader who doesn't work on P22 to have a bit more information on where the structural operons are, what is known about regulators (1 repressor controls all operons ect).
- 5) Figure S2 is used to support the notion that excision is delayed, however this was very confusing: the text (lines 162-163) indicate the integration rate was calculated as a ratio between *attL* & *attB*, but that was not clear from the legend. A simple schematic would help. Line 167 – the authors indicate changes in the integration rate, however, no replicates/error is shown, so I have no idea if this is a meaningful change (it doesn't look like it is). Rather than rely exclusively on deep sequencing approaches – which are skewed if the biological entity is circular and you are mapping to a linear integrated reference for example, I suggest the authors include some PCR based assays to monitor excision/integration. These data are simple to generate and should support analyses already included, providing the model is correct.
- 6) Given the emphasis on erroneous conclusions being drawn from lab strains/commonly used mutants (ie TS P22 undergoes ERP, no lateral transduction), it is necessary for the authors to provide some evidence that their particular strain of P22 doesn't have any mutations that could be the source of these findings (ie mutations in *Int/Xis*, repressor ect). Ensuring that natural isolates of P22 lysogens have conserved seq to the strain used here would be valuable.
- 7) The relevance of the Infection P22 panel in figure 2 is questionable and misleading. Here, there should only be episomal replication – it is unclear why the authors are plotting this as if the phage is 'part' of genome. This gives the impression the profile is something to compare the P22/ts muts to, but this is an entirely different situation.

- 8) The conclusions (and emphasis) surrounding the difference between LT by wt P22 vs the TS mutant need to be tempered to the point of only suggesting a quantitative difference in the discussion (if desired). The authors clearly demonstrate LT occurs for both, but given that both the phage & the induction condition differs in the experiments, one cannot come to the conclusions stated in this paper. For example, perhaps MMC is a more robust inducer vs the TS shift, perhaps the TS lysogen is sick owing to high levels of spontaneous induction ect. Too many variables.
- 9) Line 210 "taken together, these results show that thermal induction resulted in the classical ERP program..." this is not supported by the data, the authors show that in the absence of inducer there is already HIGH levels of expression and replication, how is one to know if that phage also 'starts' with replication then excises if the culture is already/always in that induced state?
- 10) I would suggest combining Figure 3A & 4b, (4a is not particularly informative, can be moved to the supplement, or made clearer that b is a zoom in on the relevant data). Figure 4b is extremely convincing and nicely complements the transduction assays in fig 3a.
- 11) Figure 5 – it would have been easier to follow if 4b was reported as the PFU/ml relative to an uninduced control. This would much more clearly allow the reader to follow the assay, since that would require re-doing the experiment, I instead suggest adding a schematic to help orient the reader.
- 12) A point worth considering for the discussion is that although the authors do not see a fitness cost to delayed excision (which accompanies higher LT), their assays only report 1 round of infection. Subtle fitness decreases (as seen in 5b between 60-90 min that is not yet significant) will be exacerbated by repeated passaging and thus even subtle effects would be deleterious to a phage competing with a phage that doesn't do LT in nature. It doesn't seem feasible to do such competition experiments, unless the authors have a way of comparing otherwise isogenic phages that only differ in capacity undergo LT, but their conclusions are quite strong given the nature of the experiments they are able to perform. Even their model in Fig S1 looks as if the burst for phages undergoing LT is lower! It is very counterintuitive that this would not come at a cost, perhaps the other thing to consider is the relative abundance of virions for packaging vs DNA to be packaged.
- 13) Line 314 – this statement regarding unpublished data is not sufficient, if there is no data being shown to support these conclusions this statement should not included. Along those lines however, it would be interesting for the authors to examine the position of attachment sites for phages that engage in LT in natural strains, perhaps those flanking genes could be directly linked to host fitness/supporting increased virion production ect.

Reviewer #2 (Remarks to the Author):

This manuscript by Fillol-Salom is a continuation of a previous work from this lab demonstrating that prophages can hold a different program of lytic development, involving in-situ replication (while integrated within the host chromosome), DNA packaging and lastly excision (RPE), as opposed to ERP (originally shown for phage 80-alpha of *S. aureus*). In that work they also demonstrated that RPE promotes lateral transduction. In this manuscript the authors demonstrate that this phenomenon occurs also in P22 of *Salmonella*, but in the native phage and not in the temperature sensitive mutant (which is widely used). Further they show that the RPE process allows lateral transduction without compromising the phage titer, therefore contributing to the evolution of the host. This is an important work demonstrating the differences between the two P22 phages (native vs. the temperature sensitive, *tsc229*), their induction pathways (SOS versus temperature) and their impact on lateral transduction. The experiments are nicely presented and convincing.

That said, I must say that I don't feel comfortable with the statement in the abstract and throughout the manuscript that the ERP program of the heat inducible mutant is an artefact (or the use of natural versus unnatural). One can argue that what we see in the heat sensitive mutant is the "natural" behaviour of the phage, and what we see in the native P22 upon SOS is an "adaptive" behaviour that is a result of a co-adaptation of the bacteria and the phage. Namely, it could be that the bacteria control the late excision and not the phage, and thus in respect of the phage, this is an unnatural behaviour (though eventually it benefits from it). In any case, I don't think that there are any artefacts here, but two different biological scenarios. The heat-sensitive

mutant is less adapted to its host, as due to its mutation it responds to a new mechanism of induction, which is triggered by temperature.
Besides this comment, I like the paper very much.

Additional comments that might help to improve the manuscript:

I think that the title of the paper is not informative, I recommend to change it.

Line 129 and through the manuscript: avoid the use of "unnatural".

Fig 2S and the paragraphs describing it are a bit problematic: there are no error bars in the graphs, so it is not clear how significant are the results, the y-axis does not describe integration rate, but in best excision rate, and most importantly the calculation is misleading as it is based on the attL, while it is shown that there is in situ replication which amplifies the attL as well. Thus, the calculation actually represents (excised phages)/(in-situ amplified attL-phage+originally integrated phages+ chromosomal reads). In any case, I don't think this data is essential for the manuscript.

line 194 : "delta-rep" should be "delta-pri"

line 213-5: ES18 looks to me more similar to P22ts, where both in situ replication and episomal replication proceed simultaneously.

Fig S4 is mentioned in the text after S5 and S6.

The discussion section is too short and over-simplified.

Anat Herskovits

Reviewer #3 (Remarks to the Author):

In this work Fillol-Salom et al examine and compare the induction of wild type phage P22 and a temperature sensitive mutant. They find differences in the way that the two phages enter the lytic cycle, with the wild type phage exhibiting a delayed excision and the temperature sensitive mutant phage excising early after induction. These differences result in much higher rates of lateral transduction for the wild type P22.

This work provides evidence that Salmonella phage P22 functions by a similar mechanism to the Staphylococcal phages previously characterized by the Penades lab (Science, 2018). They also provide some evidence that induction by DNA damage follows a different pathway than that induced by a temperature shift with a P22 temperature sensitive (P22-ts) mutant phage. This is a significant point to have made in the literature as phage ts mutants have been widely used in the past for phage studies and they may provide

I found the manuscript to be well written and easy to follow. One notable problem though, is that the introduction is written with a real focus on the processes that are followed by phages like P22. This would be okay if it was made clear that this is how phages like P22 work. However, the way that many parts of the paper outline the background and conclusions make it sound like all phages are like P22. There are many variations within the phage world, with some of these differences should be noted in the text. For example:

- not all temperate phages integrate into the bacterial chromosome – some are maintained as episomes
- not all phages replicate by the standard rolling circle model as implied in the introduction – phages P2 and P4 package replicated circular DNA, phage Mu transposes through the genome as it replicates and is packaged from the chromosome, ϕ -29 packages protein-terminated mature length DNA molecules.
- prophages switch to the lytic cycle for a variety of reasons, not just when their hosts begin to deteriorate. There are a number of studies that show that quorum sensing also can lead to a switch to the lytic cycle

The authors state in line 88 that prophages excise and circularize early as the first step, and that this sequence "is believed to be universal for all integrated prophages". This statement is false for several reasons, and should be removed. E. coli phage Mu does not excise and circularize, it

replicates by transposition and the genome is packaged from the bacterial chromosome. In addition, as stated later in the same paragraph, the Penades lab themselves showed that some Staphylococcal phages delay excision until later in their life cycle.

Line 119 – Many would argue that much of our current understanding of fundamental phage biology derives from studies of the classical system phage λ and other *E. coli* phages! I understand that the authors want to stress the importance of their system, but they should be careful not to oversell and to ignore or downplay the enormous body of work outside of *Salmonella* phage P22.

The authors end the introduction with the statement that “these results propose a new series of events in the life cycle of temperate prophages...” It’s not clear to me what the new series of events are – it seems that this work is confirming that wild type P22 is acting in a similar manner to the in situ replication of Staphylococcal prophages that the Penades group previously characterized (2018, *Science*). Similarly, the statement that “delayed excision leading to LT are naturally parts of the phage life cycle” also seems to imply that this was discovered in this work. It seems to me that this work confirms a similar mechanism for this particular phage, but is not a new discovery itself.

In Fig 1, the P22 *tsc229* induction time 0 (red) shows a lot of transcription. Why is this, if this timepoint is supposed to be before induction? It seems to suggest that this mutant was already spontaneously inducing before the temperature shift. When you compare this background to that seen for the wild type P22, there is a striking difference, with very little transcription observed at time 0. How can these be compared when the baseline levels are so disparate? Is there a way to decrease the levels of spontaneous induction from the P22-ts? What happens if the induction is more carefully controlled, for example, by the use of a dominant negative repressor? The worry, of course, is that some of what is being observed is because many of the cells in the P22-ts culture are already well into the lytic cycle due to the poor repression of the prophage. It’s not clear to me that the observed “early” induction from the P22-ts lysogen is not merely an artifact due to the high levels of spontaneous induction. These analyses are being done on bulk cultures, which will be a mix of silenced lysogens and inducing lysogens at any point in time.

There is a statement that there is a general leakiness of transcription and spontaneous excision observed in P22 *tsc299*. What percent of cells are spontaneously inducing? The data presented in Fig S2 appear to show that there is little change in the excision of the P22-ts mutant over the 90 minute timecourse. There is a much larger decrease noted for P22 (>0.25) as compared to P22-ts (~0.05). Why is this? How does this play into the replication cycle? How many phages are made under each of these induction conditions? Are the titers similar for the two modes of induction? The authors note that the percent of integrated P22 changed course and increased after 60 minutes, but they fail to comment on the similar increase observed for the P22-ts mutant.

Why were the infection assays to assess GT not performed for P22-ts?

The analyses of lateral transduction for P22 and P22-ts need additional context in order to appreciate the significance of the numbers presented. What are the phage titers that are released by these two methods of induction? Are they equal? The authors note that there is a high ratio of tail-defective particles resulting from the thermal induction.

In the discussion it is stated that in situ packaging initiates from some of the integrated prophages and excision occurs from some of the others. Which data specifically support these statements? Is this being proposed from previous work that should be referenced?

The statement at the end of the discussion, that the results presented here re-write important concepts of phage biology is a very strong one. The 2018 *Science* paper that first characterized this mechanism of transfer certainly described a new paradigm. It seems that the work presented here extends the original observation that Staphylococcal phages can mediate LT through late prophage excision to *Salmonella* phages. They also show that there is a difference observed with the P22-ts mutant, but because of the high levels of spontaneous induction from the prophage, the significance of the difference is difficult to pin down.

Minor points:

Figure 3 – what are metamerism spans?

Line 293-294 – phage Mu and transposable phages have been characterized in this way

Line 313 – not all pac prophages integrate into the bacterial chromosome. Wording should be changed here.

Line 314 – what are the unpublished results? Do you mean the results presented here?

Reviewer #1 (Remarks to the Author):

I do however think the paper could benefit from being re-framed: the emphasis seems to be on demonstrating that the TS P22 mutant phage is different than WT P22, the latter of which shows 'higher' levels of lateral transduction. This emphasis is likely more interesting/significant to people who study P22, but for a broad audience, this is not really a particularly compelling point. The significance is that the authors show LT in unrelated Salmonella phages – suggesting LT is widespread, and lending support to the hypothesis that it is beneficial for phages to engage in LT. For those reasons I suggest the authors more deliberately explain the relevance of including ES18 (and how it differs from P22 specifically), and include the ES18 data in the main manuscript. I would likewise suggest some of the TS P22 mutant data gets moved to the supplement.

Following the reviewer's comments, we have moved the ES18 results to the main text and now highlight the idea that LT is widespread in nature. In addition, we feel that the P22 *tsc₂29* results are important for one simple reason: when we presented the WT P22 LT results in different talks, the first thing we were always asked was, "Why wasn't this phenomenon discovered during the many decades of P22 research or in the original work on generalised or specialised transduction?" To our knowledge, none of the colleagues who posed this question to us were P22 researchers. Therefore, we feel that inclusion of the P22 *tsc₂29* data nicely addresses this recurring question.

Specific comments on figures/analysis follow below:

1) Throughout the manuscript it is not clear if replicates were done, and what exactly is being shown as far as replicates and error. This is true for all RNA and DNA-seq data (ie figs 1, 2, 4, S2 ect). The data points for each replicate should also be shown on each bar graph in addition to the error.

We thank the reviewer for the comment. For the transcriptomic analyses, as we indicate in the "Total RNA extraction and mRNA enrichment" section, "experiments were performed in triplicate". For this part, the plotting and statistical analysis were performed using the average values out of the three replicates, which provided highly consistent results (see example of bottom strand for P22 induction below).

Figure. Transcriptomic profile of P22 bottom strand after prophage induction. From left to right, different time points have been plotted for the three different replicates (each row, A, B and C). The reviewer can visually observe the high consistency across the expression profiles.

We did not include error/sd to avoid an overloaded plot, but we have now made some adjustments to account for it. First, instead of representing the coverage values for each bp, we have plotted the average for 100bp windows, as in the DNA-seq, with their respective standard deviations (see new Figure 1 in the manuscript).

For the DNA-quantification data analysis, including the results shown in Figures 2, 3, S3 and S5, there were no replicates for the different time points. In these graphs we always plotted the relative coverage (in Log₁₀ scale), calculated as absolute coverage across 100bp sliding windows along the chromosome divided by the chromosomal-non phage average coverage. We have added a sentence in “Materials and methods” to ensure the experiments performed are clear: For the DNA-quantification analyses “Only one replicate was sequenced per experiment”. This is the approach we have previously used in this type of studies. Importantly, we would like to highlight the strong consistency between experiments and time points, as observed in the DNA coverage plots for P22 Dint and P22 Dxis (Fig. 2c and d) or between phages P22 and ES18 (Fig. 2a and f, blue lines).

2) It would be helpful to orient the experiments if the authors included a time course showing PFU production for the different strains, this would help put the RNA/DNA-seq in context.

Done. See new Fig. S4.

3) It is clear from the RNA & DNA-seq data that P22 TS mutant is already induced without the temp shift, I have a hard time believing those cells are not sick and producing a significant amount of phage – this makes me concerned about suppressor mutations that could be confounding the results somewhat

The reviewer is right. The P22 *tsc29* lysogen is sick compared to the WT because of the spontaneous induction observed with this TS mutant. We have now clarified this in the text. We can also add that there are no additional mutations either in the WT or mutant phages, as confirmed by sequencing the different strains. These sequences have now been deposited and are public.

4) Figure 1 – the authors are using these data to conclude that *int/xis* are not expressed until late in infection in WT P22. However, it is unclear from the RNA-seq data alone that this is the case – this needs to be corroborated by analysis at the protein level. I also suggest that they provide a zoomed in view of the RNA-seq data on the *int/xis* operon – but again, this is not sufficient to conclude anything given that post transcriptional regulation may play a role and they are drawing their conclusions based on this data alone.

We think there is some confusion here. As indicated in the manuscript, and based in all the transcriptomic data, we proposed that the late expression of the *int* and *xis* genes occurs after P22 prophage induction, but not after P22 infection. Thus, we mentioned in the manuscript that “In contrast (to prophage induction), total viral transcripts reached maximal levels early (before 30 minutes) following infection, due to the total lack of repressor in naïve cells (Fig. 1)”.

For a more accurate characterisation of the cycle, and further to the reviewer’s comment, over the past months we have tried to quantify the amount of Xis and Int expressed either after induction of the P22 prophage or after P22 infection. To do this, we engineered strains in which the Xis and Int proteins were N-terminal or C-terminal His₍₆₎ or 3x-flag tagged, but remained under native transcriptional and translational control. While these recombinant phages and their infective particles were completely functional, we were not able to detect either the tagged Xis or Int proteins by western blot, at any of the time points analysed. This suggests that these proteins are expressed at very low levels.

As suggested by the reviewer, to better show the expression dynamics of *int/xis* at different time points, we have plotted the zoomed-in region of these genes (see new Fig. S2).

Figure S2. Zoomed-in regions for expression coverages of the *int* and *xis* genes.

I would also like to understand a little more about what is encoded downstream of *xis* that would need to be on late in infection, I'm sure there is a lot known and it would be helpful for the general reader who doesn't work on P22 to have a bit more information on where the structural operons are, what is known about regulators (1 repressor controls all operons ect).

When P22 is inserted in the bacterial chromosome, there are no additional phage-encoded genes downstream *xis* and *int* (see P22 maps in Figs. 1 and 2). When P22 excises and circularizes, the following genes correspond with the glucosyltransferase operon (*gtrA*, *gtrB* and *gtrC*), as shown below:

Figure. Region of the P22 circularized chromosome, with *int* and *xis* genes marked in red.

5) Figure S2 is used to support the notion that excision is delayed, however this was very confusing.

(i) the text (lines 162-163) indicate the integration rate was calculated as a ratio between *attL* & *attB*, but that was not clear from the legend. A simple schematic would help.

We thank the reviewer for noticing that the information provided is misleading. We have revised and corrected the description given in materials and methods (lines 534-542), results (lines 179-182) and legend of Figure S2 (now Figure S3).

For calculating the integration rates, we first counted the number of sequencing reads that spanned empty prophage attachment sites (*attB*), referred as chromosome-*attB*-chromosome reads; and the reads covering the left end of the integrated prophage (*attL* site), or chromosome-*attL*-prophage reads. Next, the number of reads spanning the *attL* site

was divided by the total number of reads, which represents the percentage of integrated prophage. As suggested by the reviewer, we have included a schematic representation of the reads used for this calculation.

Figure. Schematic representation of sequencing reads used for calculation of integration rate.

The information provided in “Materials and methods” has been updated as follows: “In addition, we counted the sequencing reads spanning the empty prophage attachment sites (*attB*) and the reads covering the left end of the integrated prophage (*attL* site). This was performed using the ‘view’ subcommand of samtools and filtering by the coordinates of the mapping reads and their respective lengths. Next, the integration percentages were obtained by dividing the reads covering the chromosome-*attL*-prophage (chr:pha) by the total reads (chr:pha + chr:chr)”.

We have rephrased the results section: “At each time point, we compared the number of sequencing reads covering the *attL* site (the left end of the integrated prophage) with the total number of reads mapping to the *attB/attL* site, which also include sequences spanning empty prophage attachment sites (*attB*). The resulting value represents the percentage of integrated prophage”.

Finally, the respective sentences of Figure S3: “The proportion of integrated phage at each timepoint was calculated by dividing the number of chromosome-*attL* reads by the total number of *attL/attB* spanning reads obtained from the sample (sum of chromosome-*attL* phage region reads plus reads spanning an unlinked chromosome-chromosome region)”.

(ii) Line 167 – the authors indicate changes in the integration rate, however, no replicates/error is shown, so I have no idea if this is a meaningful change (it doesn’t look like it is).

Following our response to comment 1, this analysis was performed using DNA-sequencing data, for which only a single replicate per experiment was available. Nevertheless, Fisher tests on the sequencing read counts (chromosome-*attB*-chromosome reads and chromosome-*attL*-prophage reads) between different time points (0 min vs 30 min, 30 min vs 60 min, and 60 min vs 90min) indicate that all apparent changes are statistically significant.

(iii) Rather than rely exclusively on deep sequencing approaches – which are skewed if the biological entity is circular and you are mapping to a linear integrated reference for example, I suggest the authors include some PCR based assays to monitor excision/integration. These

data are simple to generate and should support analyses already included, providing the model is correct.

We have not analysed whether such potential skewness would occur in our data, but it would not affect our analyses since the sequencing reads corresponding with the circular phage were not used for the calculations of the integration rate. In addition, from our experience, and since the phage can be replicating simultaneously either integrated into the bacterial chromosome (*in situ* replication) or extrachromosomally (once excised), whole-genome sequencing data at different time points provide more accurate and quantifiable results than PCR-based data.

6) Given the emphasis on erroneous conclusions being drawn from lab strains/commonly used mutants (ie TS P22 undergoes ERP, no lateral transduction), it is necessary for the authors to provide some evidence that their particular strain of P22 doesn't have any mutations that could be the source of these findings (ie mutations in *Int/Xis*, repressor ect). Ensuring that natural isolates of P22 lysogens have conserved seq to the strain used here would be valuable.

As previously mentioned, the strains in question were sequenced, including the phage ES18 which also engages in LT, and validated at the beginning of the study. The sequences for the different strains have been made available and can be downloaded from the datasets provided in this manuscript.

7) The relevance of the Infection P22 panel in figure 2 is questionable and misleading. Here, there should only be episomal replication – it is unclear why the authors are plotting this as if the phage is 'part' of genome. This gives the impression the profile is something to compare the P22/ts muts to, but this is an entirely different situation.

We apologize for any confusion this figure may have caused. We were of the opinion that this method of plotting was helpful to compare the coverage values with the rest of the experiments performed and to improve the aesthetics of the figure. As we indicated in the figure legend, we plotted the “Relative abundance of phage genomic DNA and the chromosomal regions proximal to where they integrate for P22”. To avoid any misunderstanding we have “broken up” the misleading lines, highlighted that this coverage corresponds to episomal replication, and added the coordinates where P22 is inserts into the chromosome (see figure below). Additionally, since this reviewer does not consider this figure to be essential, we have moved it to the supplementary (Fig. S5). In its place, we now include the analysis of the ES18 phage in the main figures.

8) The conclusions (and emphasis) surrounding the difference between LT by wt P22 vs the TS mutant need to be tempered to the point of only suggesting a quantitative difference in the discussion (if desired). The authors clearly demonstrate LT occurs for both, but given that both the phage & the induction condition differs in the experiments, one cannot come to the conclusions stated in this paper. For example, perhaps MMC is a more robust inducer vs the TS shift, perhaps the TS lysogen is sick owing to high levels of spontaneous induction ect. Too many variables.

Following the reviewer's comment, we have now modified the text accordingly.

9) Line 210 "taken together, these results show that thermal induction resulted in the classical ERP program..." this is not supported by the data, the authors show that in the absence of inducer there is already HIGH levels of expression and replication, how is one to know if that phage also 'starts' with replication then excises if the culture is already/always in that induced state?

Corrected.

10) I would suggest combining Figure 3A & 4b, (4a is not particularly informative, can be moved to the supplement, or made clearer that b is a zoom in on the relevant data). Figure 4b is extremely convincing and nicely complements the transduction assays in fig 3a.

We have combined the figures as suggested. However, we have kept Fig. 4a as it provides a big picture of the chromosomal DNA that is packaged after prophage induction. We have also included the ES18 data in this figure.

11) Figure 5 – it would have been easier to follow if 4b was reported as the PFU/ml relative to an uninduced control. This would much more clearly allow the reader to follow the assay, since that would require re-doing the experiment, I instead suggest adding a schematic to help orient the reader.

Although we had that data, we did not include it in the initial figure, assuming that it was unnecessary. Following the reviewer's comment, we have created a new figure including the uninduced control.

12) A point worth considering for the discussion is that although the authors do not see a fitness cost to delayed excision (which accompanies higher LT), their assays only report 1 round of infection. Subtle fitness decreases (as seen in 5b between 60-90 min that is not yet significant) will be exacerbated by repeated passaging and thus even subtle effects would be deleterious to a phage competing with a phage that doesn't do LT in nature. It doesn't seem feasible to do such competition experiments, unless the authors have a way of comparing otherwise isogenic phages that only differ in capacity undergo LT, but their conclusions are quite strong given the nature of the experiments they are able to perform. Even their model in Fig S1 looks as if the burst for phages undergoing LT is lower! It is very counterintuitive that this would not come at a cost, perhaps the other thing to consider is the relative abundance of virions for packaging vs DNA to be packaged.

We agree that early excision would be beneficial in terms of phage production but comes at the expense of being detrimental to gene transfer. What we propose is that prophages such as P22 have found an interesting compromise between these two processes, delaying excision to engage in LT. We have clarified this now in the discussion.

13) Line 314 – this statement regarding unpublished data is not sufficient, if there is no data being shown to support these conclusions this statement should not included. Along those lines however, it would be interesting for the authors to examine the position of attachment sites for phages that engage in LT in natural strains, perhaps those flanking genes could be directly linked to host fitness/supporting increased virion production ect.

Following the reviewer's comment, the text has been modified accordingly. The reviewer is also right about the type of genes that are located next to the *attB* sites. We demonstrated, in our original paper describing LT in *S. aureus*, that many pathogenicity islands are located there and can be mobilised by LT. We have now demonstrated elsewhere that there is a similar organisation in *Salmonella* (paper in revision in Nature Communications). This information has been now included and referenced in the new discussion.

Reviewer #2 (Remarks to the Author):

That said, I must say that I don't feel comfortable with the statement in the abstract and throughout the manuscript that the ERP program of the heat inducible mutant is an artefact (or the use of natural versus unnatural). One can argue that what we see in the heat sensitive mutant is the "natural" behaviour of the phage, and what we see in the native P22 upon SOS is an "adaptive" behaviour that is a result of a co-adaptation of the bacteria and the phage. Namely, it could be that the bacteria control the late excision and not the phage, and thus in respect of the phage, this is an unnatural behaviour (though eventually it benefits from it). In any case, I don't think that there are any artefacts here, but two different biological scenarios. The heat-sensitive mutant is less adapted to its host, as due to its mutation it responds to a new mechanism of induction, which is triggered by temperature. Besides this comment, I like the paper very much.

We thank this reviewer for her nice comments about the paper.

The reviewer's points on unnatural vs natural are well taken. Our use of the terms "mutant" or "unnatural" are simply arbitrary designations and refer to the fact that the provenance of the P22 *tsc₂29* phage is known, as it was generated in the laboratory and was derived from what the literature refers to as "WT" P22. Therefore, we refer to P22 *tsc₂29* phage as a mutant of P22 and we consider WT P22 to be the natural phage. As indicated now in the text, the TS prophage is highly costly for its host because it causes a growth defect, suggesting that prophages carrying this type of mutation would not persist and fare well in nature. In any case, we have modified the text in some parts of the manuscript to avoid the unnecessary use of these words; however, we have kept it in some parts while making it clear why we use this terminology.

Additional comments that might help to improve the manuscript:

I think that the title of the paper is not informative, I recommend to change it.

Following the reviewer's comment, we have now modified the title (Lateral transduction is inherent to the life cycle of the archetypical *Salmonella* phage P22). We hope this is an improvement.

Line 129 and through the manuscript: avoid the use of "unnatural".

See the previous comment.

Fig 2S and the paragraphs describing it are a bit problematic: there are no error bars in the graphs, so it is not clear how significant are the results, the y-axis does not describe integration rate, but in best excision rate, and most importantly the calculation is misleading as it is based on the attL, while it is shown that there is in situ replication which amplifies the attL as well. Thus, the calculation actually represents (excised phages)/(in-situ amplified attL-phage+originally integrated phages+ chromosomal reads). In any case, I don't think this data is essential for the manuscript.

We acknowledge that the description of the calculation of the integration rates was misleading, with erroneous information provided in methods and lack of clarity in the results section. The integration rates were calculated by counting the number of sequencing reads that spanned empty prophage attachment sites (*attB*, ie chromosome-*attB*-chromosome) and the reads covering the left end of the integrated prophage (*attL* site, ie chromosome-*attL*-prophage). Next, the number of reads spanning the *attL* site was divided by the total number of reads, which represents the percentage of integrated prophage. Since we did not use the reads corresponding only with phages, the calculation does not represent excised phage. For more details, we would like to refer the reviewer to the response given previously to reviewer 1.

This figure did not include error bars because the analysis relied on DNA-sequencing data, for which only a single replicate per experiment was available. However, we would like to highlight the consistency between P22 and ES18 prophages, and with the results from our study on lateral transduction, where we first described this event (Chen et al., 2018). Since we believe this data is important to help understanding that prophages excise late from the bacterial chromosome, we maintained it as a supplementary figure.

line 194 : "delta-rep" should be "delta-pri"

Corrected.

line 213-5: ES18 looks to me more similar to P22ts, where both in situ replication and episomal replication proceed simultaneously.

We agree with the reviewer that both in ES18, like in P22, in situ replication and episomal replication occurs simultaneously. This is, however, different to what is seen with the P22ts, where the episomal replication occurs even before induction.

Fig S4 is mentioned in the text after S5 and S6.

Corrected.

The discussion section is too short and over-simplified.

We have now extended the discussion to include some of the comments made by the reviewers.

Reviewer #3 (Remarks to the Author):

I found the manuscript to be well written and easy to follow. One notable problem though, is that the introduction is written with a real focus on the processes that are followed by phages like P22. This would be okay if it was made clear that this is how phages like P22 work. However, the way that many parts of the paper outline the background and conclusions make it sound like all phages are like P22. There are many variations within the phage world, with some of these differences should be noted in the text.

Following the reviewer's suggestion, we have edited the introduction to make it clear that the paper is related to phages that are similar to P22.

For example:

- not all temperate phages integrate into the bacterial chromosome – some are maintained as episomes.

Corrected.

- not all phages replicate by the standard rolling circle model as implied in the introduction – phages P2 and P4 package replicated circular DNA, phage Mu transposes through the genome as it replicates and is packaged from the chromosome, ϕ -29 packages protein-terminated mature length DNA molecules.

Corrected.

- prophages switch to the lytic cycle for a variety of reasons, not just when their hosts begin to deteriorate. There are a number of studies that show that quorum sensing also can lead to a switch to the lytic cycle.

Corrected.

The authors state in line 88 that prophages excise and circularize early as the first step, and that this sequence “is believed to be universal for all integrated prophages”. This statement is false for several reasons, and should be removed. E. coli phage Mu does not excise and circularize, it replicates by transposition and the genome is packaged from the bacterial chromosome. In addition, as stated later in the same paragraph, the Penades lab themselves showed that some Staphylococcal phages delay excision until later in their life cycle.

Corrected, as suggested by the reviewer.

Line 119 – Many would argue that much of our current understanding of fundamental phage biology derives from studies of the classical system phage λ and other E. coli phages! I understand that the authors want to stress the importance of their system, but they should be careful not to oversell and to ignore or downplay the enormous body of work outside of Salmonella phage P22.

The reviewer is completely right, and the manuscript has been corrected accordingly,

The authors end the introduction with the statement that “these results propose a new series of events in the life cycle of temperate prophages...” It’s not clear to me what the new series of events are – it seems that this work is confirming that wild type P22 is acting in a similar manner to the in situ replication of Staphylococcal prophages that the Penades group previously characterized (2018, Science). Similarly, the statement that “delayed excision leading to LT are naturally parts of the phage life cycle” also seems to imply that this was discovered in this work. It seems to me that this work confirms a similar mechanism for this particular phage, but is not a new discovery itself.

Corrected.

In Fig 1, the P22 tsc229 induction time 0 (red) shows a lot of transcription. Why is this, if this timepoint is supposed to be before induction? It seems to suggest that this mutant was already spontaneously inducing before the temperature shift. When you compare this background to that seen for the wild type P22, there is a striking difference, with very little transcription observed at time 0. How can these be compared when the baseline levels are so disparate?

The reviewer’s description of the “leakiness” of the P22 tsc₂₂₉ mutant is accurate, and precisely why we reported these findings. The point of the comparison was to show that the P22 tsc₂₂₉ mutant does not behave like WT and shows a completely different life cycle. We propose that the P22 tsc₂₂₉ repressor does not repress as well as the WT repressor under normal conditions. Furthermore, the TS repressor is inactivated (nearly instantaneously) by a temperature shift that likely unfolds the protein, rather than by the normal mechanism which is self-cleavage by activated RecA during the SOS response. This in turn results in high expression from the beginning and rapid de-repression, which we suggest was the origins of the model that proposed that excision of the P22 prophage occurs early in its life cycle immediately after prophage induction.

Is there a way to decrease the levels of spontaneous induction from the P22-ts? What happens if the induction is more carefully controlled, for example, by the use of a dominant negative repressor? The worry, of course, is that some of what is being observed is because many of the cells in the P22-ts culture are already well into the lytic cycle due to the poor repression of the prophage. It's not clear to me that the observed "early" induction from the P22-ts lysogen is not merely an artifact due to the high levels of spontaneous induction. These analyses are being done on bulk cultures, which will be a mix of silenced lysogens and inducing lysogens at any point in time.

The reviewer brings up valid concerns, and had we set out to use the P22 *tsc₂29* mutant for a temperature shift system to study P22, we would have certainly looked to control the levels of spontaneous induction. However, in this study, our intention was to use the same strains and repeat the same induction methods that were used in previous publications to understand why LT was not discovered with P22. Unfortunately, the P22 *tsc₂29* mutant and the methods used for induction result in high levels of spontaneous induction. We have also reverted the mutation in the *c2* repressor and the repaired prophage behaves the same as WT P22. Therefore, the single C59R mutation is clearly the cause of the P22 *tsc₂29* spontaneous induction and the origin of its different behaviour.

There is a statement that there is a general leakiness of transcription and spontaneous excision observed in P22 *tsc₂29*. What percent of cells are spontaneously inducing? The data presented in Fig S2 appear to show that there is little change in the excision of the P22-ts mutant over the 90 minute timecourse. There is a much larger decrease noted for P22 (>0.25) as compared to P22-ts (~0.05). Why is this? How does this play into the replication cycle? How many phages are made under each of these induction conditions? Are the titers similar for the two modes of induction? The authors note that the percent of integrated P22 changed course and increased after 60 minutes, but they fail to comment on the similar increase observed for the P22-ts mutant.

We have not quantified the percentage of cells that are spontaneously induced. We have now included a new figure (Figure S4) showing the number of phage particles that are produced from each of the different prophages after induction. As can be seen in that Figure, the titre obtained after induction of the P22 *tsc₂29* mutant is higher than that observed for WT P22, while the transfer of the markers by LT is proportionally higher in the WT phage. As previously mentioned, the different behaviour observed between the WT and the *tsc₂29* phages is in part due to the high expression of the *xis* and *int* genes that occurs in the latter, even in absence of prophage induction. The extremely rapid inactivation of the *tsc₂29* repressor by a temperature shift that unfolds the TS protein also plays a role. As can be seen in the Figure 2b, which analyses prophage replication, that implies that even before induction, most of the P22 *tsc₂29* mutant replicates episomally, while WT P22 starts replicating *in situ* after induction. These differences explain why the titre of the P22 *tsc₂29* mutant is higher than that observed for the WT, while the WT phage is better at engaging in LT compared to the TS mutant.

Why were the infection assays to assess GT not performed for P22-ts?

This data has been now included in the text (Fig. S8).

The analyses of lateral transduction for P22 and P22-ts need additional context in order to appreciate the significance of the numbers presented. What are the phage titers that are released by these two methods of induction? Are they equal? The authors note that there is a high ratio of tail-defective particles resulting from the thermal induction.

As previously mentioned, the titre for the P22 *tsc₂29* mutant was higher than that observed for the WT phage. All of these numbers have been now included in the Source Data file and in the new Figure S4. We have not specifically analysed the presence of the tail-defective particles after induction of the prophages. This refers to old observations made when the TS

mutant was analysed. To avoid confusion, this sentence has been now removed from the text.

In the discussion it is stated that *in situ* packaging initiates from some of the integrated prophages and excision occurs from some of the others. Which data specifically support these statements? Is this being proposed from previous work that should be referenced?

We proposed this idea in our 2018 paper. That is the only hypothesis that fits well with the obtained results. To get functional infective particles, the excision must occur from prophages that have not initiated *in situ* packaging. Otherwise, once TerS cuts the prophage to initiate packaging, the prophage will not be able to generate functional phage particles since its genome has been split in two parts.

The statement at the end of the discussion, that the results presented here re-write important concepts of phage biology is a very strong one. The 2018 Science paper that first characterized this mechanism of transfer certainly described a new paradigm. It seems that the work presented here extends the original observation that Staphylococcal phages can mediate LT through late prophage excision to Salmonella phages. They also show that there is a difference observed with the P22-ts mutant, but because of the high levels of spontaneous induction from the prophage, the significance of the difference is difficult to pin down.

This sentence has been modified in response to the reviewer's comments.

Minor points:

Line 293-294 – phage Mu and transposable phages have been characterized in this way.

Corrected.

Line 313 – not all pac prophages integrate into the bacterial chromosome. Wording should be changed here.

Modified.

Line 314 – what are the unpublished results? Do you mean the results presented here?

The unpublished results were those saying that phages benefit from LT. We have now modified the text to express this in a different way.

REVIEWER COMMENTS

Reviewer #1 (Remarks to the Author):

I agree with Reviewer 2 regarding use of the term 'unnatural'. I suggest the authors use 'artificial' (or artifactual) in place of unnatural as in lines 134, 141, 184, 230.

As per my original review - the schematic provided in the rebuttal for Fig S3 was not included in the revised figure. Again the description of the calculation in the text / legend is confusing; in the main text it appears the calculation would be: $\text{attL}/(\text{attL}+\text{attB})$; from the schematic perhaps this is correct, though the legend says 'sum of attL plus reads spanning an unlinked chromosome-chromosome region' - is the unlinked chromosome-chromosome region attB or just an arbitrary location?

I am also disappointed that the authors are drawing these conclusions from N=1 sample in these data without any additional corroborating evidence (such as protein levels or PCR verification for the circularized junction as suggested in my first review) and that they justify this approach because reviewers either missed it or otherwise excused it in a previous publication. I do not agree that this overstep should persist.

Minor:

Line 73 - additional is misspelled

line 82 add the word undergo: 'are also induce to undergo self-cleavage by activated RecA.'

Line 160: They introduce the C2 C59R mutant but don't use this term again in the paper. is this the temperature sensitive mutant?

Reviewer #2 (Remarks to the Author):

The authors addressed my comments.

I accept the paper.

Reviewer #3 (Remarks to the Author):

My comments have been adequately addressed.

Reviewer #1 (Remarks to the Author):

I agree with Reviewer 2 regarding use of the term 'unnatural'. I suggest the authors use 'artificial' (or artifactual) in place of unnatural as in lines 134, 141, 184, 230.

Corrected.

As per my original review - the schematic provided in the rebuttal for Fig S3 was not included in the revised figure. Again the description of the calculation in the text / legend is confusing; in the main text it appears the calculation would be: $\text{attL}/(\text{attL}+\text{attB})$; from the schematic perhaps this is correct, though the legend says 'sum of attL plus reads spanning an unlinked chromosome-chromosome region' - is the unlinked chromosome-chromosome region attB or just an arbitrary location?

Following the reviewer's suggestion, we have now included the schematic figure (see new Figure S3a). We have also corrected the figure legend.

I am also disappointed that the authors are drawing these conclusions from N=1 sample in these data without any additional corroborating evidence (such as protein levels or PCR verification for the circularized junction as suggested in my first review) and that they justify this approach because reviewers either missed it or otherwise excused it in a previous publication. I do not agree that this overstep should persist.

Following the reviewer's suggestion, we have now validated all our results using qPCR (see new Fig. S6). We have obtained 3 replicates of each prophage (after induction), at different times points, and have analysed both prophage excision and circularisation.

Minor:

Line 73 - additional is misspelled

Corrected.

line 82 add the word undergo: 'are also induce to undergo self-cleavage by activated RecA.'

Corrected.

Line 160: They introduce the C2 C59R mutant but don't use this term again in the paper. is this the temperature sensitive mutant?

Yes, it is. Now clarified.

Reviewer #2 (Remarks to the Author):

The authors addressed my comments. I accept the paper.

Thanks for your support.

Reviewer #3 (Remarks to the Author):

My comments have been adequately addressed.

Thanks for your support.

REVIEWERS' COMMENTS

Reviewer #1 (Remarks to the Author):

My comments have been adequately addressed.

Reviewer #1 (Remarks to the Author):

My comments have been adequately addressed.

Thanks for your support.